# Structural and Biochemical Characterization of Silver/Copper Binding by *Dendrorhynchus zhejiangensis* Ferritin

**DOI:** 10.3390/polym15051297

**Published:** 2023-03-03

**Authors:** Chunheng Huo, Tinghong Ming, Yan Wu, Hengshang Huan, Xiaoting Qiu, Chenyang Lu, Ye Li, Zhen Zhang, Jiaojiao Han, Xiurong Su

**Affiliations:** 1State Key Laboratory for Managing Biotic and Chemical Threats to the Quality and Safety of Agro-Products, Ningbo University, Ningbo 315211, China; 2School of Marine Science, Ningbo University, Ningbo 315832, China; 3Key Laboratory of Aquacultural Biotechnology Ministry of Education, Ningbo University, Ningbo 315832, China; 4College of Food and Pharmaceutical Sciences, Ningbo University, Ningbo 315832, China

**Keywords:** *Dendrorhynchus zhejiangensis*, ferritin, marine invertebrate, Ag^+^, Cu^2+^, iron-binding capacity

## Abstract

Ferritin with a highly symmetrical cage-like structure is not only key in the reversible storage of iron in efficient ferroxidase activity; it also provides unique coordination environments for the conjugation of heavy metal ions other than those associated with iron. However, research regarding the effect of these bound heavy metal ions on ferritin is scarce. In the present study, we prepared a marine invertebrate ferritin from *Dendrorhynchus zhejiangensis* (DzFer) and found that it could withstand extreme pH fluctuation. We then demonstrated its capacity to interact with Ag^+^ or Cu^2+^ ions using various biochemical and spectroscopic methods and X-ray crystallography. Structural and biochemical analyses revealed that both Ag^+^ and Cu^2+^ were able to bind to the DzFer cage via metal-coordination bonds and that their binding sites were mainly located inside the three-fold channel of DzFer. Furthermore, Ag^+^ was shown to have a higher selectivity for sulfur-containing amino acid residues and appeared to bind preferentially at the ferroxidase site of DzFer as compared with Cu^2+^. Thus, it is far more likely to inhibit the ferroxidase activity of DzFer. The results provide new insights into the effect of heavy metal ions on the iron-binding capacity of a marine invertebrate ferritin.

## 1. Introduction

Transition-metal ions are vital components of terrestrial life; however, the intrinsic reactivity of these aqua ions, whether as Lewis acid catalysts or redox catalysts, is a challenge for cells committed to physiological metal homeostasis [1]. In particular, both iron deficiency and iron overload can have extremely deleterious consequences for iron homeostasis. Thus, the importance of ferritin in the strict regulation of the iron concentration in living cells cannot be overemphasized [2]. It has been demonstrated that ferritin is a class of well-conserved iron storage and detoxification proteins commonly found in the cytosol of both prokaryotes and eukaryotes [3,4]. The majority of ferritins consist of 24 subunits assembled in a highly symmetric manner. They compose a spherical protein shell around a central cavity into which a ferric oxyhydroxide mineral core can be formed [4]. In vertebrates, ferritins are generally composed of heteropolymers of two types of three subunits, i.e., heavy (H), middle (M), and light (L) subunits [5]. The H subunit is similar to both the M subunit (~85% sequence identity) and the L subunit (~55% sequence similarity), but their functions are discrete [1,6]. The H and M subunits mainly function in the rapid oxidation of ferrous ions (Fe^2+^) at the ferroxidase center, whereas the M and L subunits are responsible for facilitating iron mineralization and the stabilization of the iron mineral core [7,8]. It is relevant to note that ferritin can take up their substrate Fe^2+^ ions from solution, oxidize them, and store these ferric minerals inside the 24-mer shell-like structure, accommodating up to 2000~3000 iron atoms [9]. Although many biochemical studies focus on the performance of iron-loaded ferritins, the corresponding mechanistic details of iron uptake and transportation in ferritins from marine invertebrates remain to be elucidated.

Structurally, a canonical ferritin subunit is usually composed of four *α*-helix bundles (A, B, C, D) with a ferroxidase center and a short helix (E) at a 60° angle toward the C-terminal [7,10]. As is well known, these subunits are reversibly assembled into a hollow nanocage with a diameter of ~12 nm and a 4-3-2 symmetrical architecture, resulting in six four-fold, eight three-fold and twelve two-fold channels [4,5]. In spite of their small sizes (a diameter of 0.2~0.5 nm and a length of 3~4 nm), these channels can act as a bridge between the internal cavity and the external environment to pump Fe^2+^, oxygen, and other molecules in and out of ferritin [5,11]. Furthermore, numerous experiments show that the hydrophilic three-fold channels in ferritins also have acknowledged the potential gates for the entry of non-native metal ions (other than Fe^2+^) into the cage, similar to metal ion transport in transmembrane ion channels [5,12]. Although the accumulation of iron ions in the ferritin cage has been systematically and extensively explored, to date, there are only a limited number of detailed reports on the accumulation of non-natural transition metal ions in the ferritin cage.

Previous research has revealed that transition metal ions, such as Cu^2+^ and Ag^+^, have high binding affinities for most amino acids in ferritins. They can generate much stronger coordination forces between them than intermolecular forces (i.e., hydrogen bonds and van der Waals forces) [12,13,14]. It is also reported that copper ions (Cu^2+^) are bivalent transition metal ions that are not prone to oxidation. One of their coordination properties is the ability to deprotonate and interact with amide nitrogen atoms of the His residue to form high-stability chelates [15]. For instance, it was demonstrated that Cu^2+^ had a binding affinity to ferroxidase sites to inhibit iron binding, which could indirectly reduce ferroxidase activity [12,14]. Moreover, Cu^2+^ also exhibits a preferentially binding function in coordination with the sulfur atom of the Cys residue, and it can additionally bind to the three-fold channel of frog M ferritin (FrMF) (with the exception of the ferroxidase center) to form a thiol bridge (Cys) structure supported by His and Glu [14,16]. For comparison, silver ions (Ag^+^) are universally acknowledged as essential noble metal ions and are used as the basis of nanomaterial formations [17]. It is noteworthy that ferritin’s natural hydrophilic three-fold channels and multi-ion chelation centers provide both a potential gateway for the entry of metal ions such as Ag^+^ and a well-defined coordination environment for protein-metal interactions [17,18,19]. Consequently, Kasyutich et al. revealed the presence of specific binding and nucleation sites of Ag^+^ in *Pyrococcus furiosus* ferritin (PfFt), suggesting that these silver binding sites were not conserved in other ferritin templates [18]. Nonetheless, very few studies have explored the structural basis of Cu^2+^ and Ag^+^ binding to ferritins from marine invertebrates.

As a benthic marine invertebrate, *Dendrorhynchus zhejiangensis* was first found at the bottom of shrimp ponds along the eastern coast of Fenghua city in Zhejiang province. Inspired by the living condition of *D. zhejiangensis*, which usually inhabits heavy metal-rich environments [20], we aim to establish whether *D. zhejiangensis* ferritin (DzFer) molecules have the capability to accumulate heavy metal ions such as Ag^+^ or Cu^2+^. In this study, using biochemical experimentation, we demonstrated that DzFer has the ability to interact with Ag^+^ or Cu^2+^. We also demonstrated the structural basis of the binding sites of Ag^+^-bound DzFer (Ag^+^-DzFer) and Cu^2+^-bound DzFer (Cu^2+^-DzFer) using X-ray crystallographic data. The present study increases our fundamental understanding of specific metal–protein interactions.

## 2. Material and Methods

### 2.1. Protein Expression and Purification

As previously reported by Huan et al. [21], SUMO-tagged DzFer was obtained via low-temperature-induced expression and purification with a Ni–NTA agarose (GE Healthcare, Chicago, IL, USA) column pre-equilibrated with binding buffer (25 mM Tris–HCl pH 8.0, 150 mM NaCl). For the purification of tag-free DzFer, the protein was mixed with nine parts of binding buffer and then incubated with SUMO protease and protein at a ratio of 1:500 overnight at 4 °C. After the removal of the SUMO tag, the flowthrough was concentrated to 20 mL (approx. 1.0 mg/mL) using a 30 kDa molecular weight cutoff (MWCO) Amicon Ultra-4 centrifugal filter (Merck Millipore, Billerica, MA, USA) and then further purified with a HiLoad 16/600 Superdex 200 pg gel-filtration chromatography column (GE Healthcare) in binding buffer. Protein concentrations were determined using bovine serum albumin (BSA) as standard and a Bradford Protein Assay Kit (Beyotime Institute of Biotechnology, Shanghai, China). The color of the purified DzFer presented yellow (5 mg/mL) and light (1 mg/mL), respectively (Appendix A). The protein solution was concentrated to 20 mg/mL and then flashed frozen with liquid nitrogen and stored at −80 °C.

### 2.2. Polyacrylamide Gel Electrophoresis (PAGE) Analysis

The presence and purity of DzFer were visually assessed using SDS-PAGE and Native-PAGE. For the SDS-PAGE experiment, protein samples were mixed with an equal volume of 5 × loading buffer containing DTT and SDS and heated at 100 °C for 10 min in a dry heating block. The samples were then centrifuged at maximum speed for 5 min using a bench-top centrifuge. Subsequently, 10 μL of protein samples (~5 μg of protein) were loaded into each well. The electrophoresis of DzFer under denaturing conditions was conducted using 12% SDS-PAGE, continuing at 120 V for approx. 2.5 h. For Native-PAGE, native polyacrylamide gels were obtained using a Native PAGE Preparation Kit (Sangon Biotech Co., Ltd., Shanghai, China). Protein samples were placed in 6% polyacrylamide gradient gels at 120 V for approx. 3 h at 4 °C, employing 125 mM Tris–HCl (pH 8.3) and 1.25 M Glycine as the running buffer. Then, proteins were visualized by staining with Coomassie Brilliant Blue R-250.

### 2.3. pH Stability Analysis

Different pH solutions were prepared by using 25 mM Tris–HCl buffer containing 150 mM NaCl adjusted to the different pH values (pH 2, 4, 6, 8, 10, and 12) with the addition of small amounts of hydrochloric acid or sodium hydroxide through a Mettler Toledo FE28-Standard pH meter according to the previously reported method [22]. The DzFer sample was diluted to approx. 0.5 mg/mL with binding buffer and incubated overnight in different pH buffers. Then, the protein samples were dialyzed using dialysis tubing with a cutoff of 15 kDa MWCO (Merck Millipore) against various pH condition buffers for 3 h at room temperature under low-frequency magnetic stirring. The corresponding pH buffer was replaced every hour. The resulting protein samples were collected for the subsequent experiments.

### 2.4. Ag^+^/Cu^2+^-Binding Assay

The DzFer sample was diluted to approx. 0.1 mg/mL using 50 mM Tris (pH 8.0) with a total volume of 200 mL. For the preparation of Ag^+^-DzFer, 100 mL portions of the protein sample were dialyzed against 2 L of solution containing 0.4 mM AgNO_3_ at 18 °C overnight under magnetic stirring, as previously described [17]. For Cu^2+^-DzFer, another 100 mL of the protein sample was dialyzed against 2 L of solution containing 2.5 mM CuCl_2_ at 18 °C overnight under magnetic stirring, as previously described [10,23]. Thereafter, the resulting sample was dialyzed in the buffer solution. During this time, the solution was replaced 3–4 times to remove free Cu^2+^ ions. Excess metal ions were removed by dialysis against 50 mM Tris pH 8.0. Subsequently, both Ag^+^-DzFer and Cu^2+^-DzFer were concentrated using dialysis tubing with a cutoff of 30 kDa MWCO (Merck Millipore) and stored at 4 °C. The method for measuring the concentrations of protein samples was as described above.

### 2.5. Biochemical Characterization Analysis

#### 2.5.1. Transmission Electron Microscopy (TEM)

TEM characterization was performed using a Hitachi H-7650 transmission electron microscope (Hitachi Co., Ltd., Tokyo, Japan) at 80 kV. Protein samples of TEM were diluted to approx. 0.25 mg/mL with binding buffer, and 10 μL of these samples was placed onto copper grids with a carbon coating. After being adsorbed for 2 min, the uranyl acetate (approx. 2%) was used to stain the samples for 30 s. Each sample was observed in more than five regions to avoid experimental errors.

#### 2.5.2. Circular Dichroism (CD)

The concentration of protein samples was adjusted to approx. 0.2 mg/mL in 10 mM sodium phosphate (pH 7.4). As previously described [23,24], CD spectra were collected using a Jasco J-1500 CD Spectropolarimeter (JASCO Corp., Tokyo, Japan) in the wavelength range from 190 nm to 260 nm at 25 °C, and the scan rate was 1 nm/min. Samples were scanned three times, and then an average was taken with a 1.0 nm bandwidth. All spectra were recorded using a quartz cell with 1 cm path length. The wavelengths at which the high-tension (HT) voltage exceeded 700 V were excluded during the CD measurements. The blank was the spectrum of deionized water. The proportions of the *α*-helix, *β*-turn, *β*-sheet, and the random coil were estimated using Yang’s equation software.

#### 2.5.3. Dynamic Light Scattering (DLS)

Prior to detection for the DLS analysis, the concentrations of DzFer, Ag^+^-DzFer, and Cu^2+^-DzFer were centrifuged at 12,000× *g* for 10 min at room temperature and subsequently adjusted to approx. 0.25 mg/mL with 50 mM Tris (pH 8.0) buffer. The size distribution analysis of protein samples was performed using a Zetasizer Nano ZS instrument (Malvern Instruments Ltd., UK) at 25 °C. The hydrodynamic radius (R_H_) distributions of the prepared samples were calculated using the OmniSIZE 3.0 software.

#### 2.5.4. Inductively Coupled Plasma Mass Spectrometry (ICP-MS)

The metal contents of the protein samples were determined using a Thermo X Series II ICP-MS instrument (Thermo Fisher Scientific Inc., MA, USA), as previously reported [21]. The protein concentrations and metal contents were used to calculate the average number of metal atoms loaded per ferritin cage [25], as reported by Si et al. All samples were assessed at least twice.

#### 2.5.5. Spectrophotometric Characterization

The ultraviolet–visible (UV–vis) absorption spectral measurements were performed using an Epoch 2 microplate spectrophotometer (BioTek Instruments, Inc., Winooski, VT, USA) in the range of 200–600 nm. The absorbance spectra were recorded using the protein concentrations of 3, 1.5, 0.75, and 0.375 mg/mL, respectively, in binding buffer at room temperature. The experimental conditions were as follows: bandwidth 1.0 nm; scanning speed 100 nm/min; data pitch 1.0 nm.

### 2.6. Crystallization, Data Collection, and Structure Determination

The native crystals of DzFer were grown by mixing 1 μL (approx. 16 mg/mL in binding buffer) of protein sample with 1 μL of reservoir solution (0.2 M ammonium sulfate, 0.1 M sodium acetate trihydrate pH 4.6, 30% *w*/*v* PEG monomethyl ether 2000) (condition I), equilibrating against 150 μL of the reservoir solution (Appendix A). For the soaking method, DzFer crystals grown from condition I were soaked for 1 min in the condition I solution supplemented with 20% glycerol and 100 mM AgNO_3_. The crystals soaking silver ions were then washed with the condition I solution containing 20% glycerol to remove the unbound ions. For Cu^2+^-DzFer crystals, the Cu^2+^-DzFer sample was firstly concentrated to approx. 10 mg/mL for crystallization using an Amicon Ultra-15 centrifugal filter (MWCO: 30 kDa) (Merck Millipore). The initial screening of crystallization conditions was performed using the sitting-drop vapor diffusion method in 96 well-plates using the Crystal Screen kits Ⅰ & Ⅱ (Hampton Research Corp., CA, USA) at 18 °C. After optimizing the crystallization conditions, these well-diffracting crystals were obtained in a droplet containing 1.0 μL of protein solution (approx. 5.5 mg/mL in 100 mM MES, pH 5.5) and 1.0 μL of reservoir solution (0.2 M sodium citrate tribasic dihydrate, 0.1 M Tris hydrochloride pH 8.5, 30% *v*/*v* polyethylene glycol 400) equilibrating against 150 μL of reservoir solution (Appendix A). All crystals for data collection were soaked in the reservoir solution supplemented with 20% glycerol as a cryoprotectant, followed by flash freezing in liquid nitrogen for storage.

All the X-ray diffraction data sets were collected under cryogenic conditions (100 K) on beamlines BL02U1 and BL18U1 at the Shanghai Synchrotron Radiation Facility (SSRF) [26]. Data were processed with indexing, integrating, and scaling using the XDS program package [27] and the CCP4 suite [28]. The dataset was submitted to the BALBES server for model building [29]. The initial phases for each structure were determined by molecular replacement with DzFer (PDB ID: 7EMK) [21] and FrMF (PDB ID: 3KA8) [30] as the search models. Structure refinement was performed using the *phenix.refine* program in PHENIX [31] and REFMAC5 in CCP4 suite [32], and manual model adjustments were made with COOT [33] on the basis of *σ*-weighted (2*Fo*–*Fc*) and (*Fo*–*Fc*) electron density maps. Water molecules were added using the built-in find-water function of COOT according to a lower cutoff of 3*σ* in the *Fo*–*Fc* map [34] and individually checked for a significant signal and consistent contact with the H-bond donor/acceptor [35]. Metal ions were located in the strong difference peaks (*Fo*–*Fc*) map and then confirmed based on shorter bond distances with neighboring water molecules or other protein residual ligands [34]. Non-crystallographic symmetry restraints were not applied, while the metal positions were restrained by refinement with the *phenix.refine* program [31]. Metal ion coordination geometry was further verified by applying the CheckMyMetal server [36]. The quality validations of the overall structures were conducted using MolProbity [37]. The space group and crystallographic origin validation was performed using the Zanuda program [38]. The superimposed structures were performed using the Superpose version 1.0 (http://superpose.wishartlab.com/, accessed on 23 August 2022) [39]. Molecular graphics renditions for all protein structures were generated using PyMOL [40]. Data collection and structural refinement statistics are presented in Appendix A. The secondary structure assignments from protein structures were conducted using the STRIDE server (http://webclu.bio.wzw.tum.de/stride/, accessed on 13 February 2023) [41].

### 2.7. Iron Oxidation Assay

In order to test the oxidation reactions of protein samples toward Fe^2+^, a ferroxidase assay was performed based on the approach in [42], as described by Khare et al. For specifically measuring Fe^3+^, a fresh solution of 100 mM HEPES (pH 6.5) containing 256 µM of FeSO_4_⸱7H_2_O was mixed with a protein solution of 0.5 µM (25 µL). The kinetic experiments were monitored using a Varioskan LUX multimode microplate reader (Thermo Fisher Scientific) with an optical density of 310 nm from the time of iron addition (a total of 200 µL). Ferrozine is regarded as a colorimetric reagent and is commonly used to determine oxidized iron content. This is possible because ferrozine generates a violet-colored Fe^2+^–ferrozine complex with Fe^2+^ ion in solution, which can be detected at different time points, e.g., 0, 15, 30, 60, 90, and 120 s at 560 nm [43]. The iron oxidation kinetics of DzFer, Ag^+^-DzFer, and Cu^2+^-DzFer were assessed using an Epoch 2 microplate spectrophotometer (BioTek Instrument, Inc., Winooski, VT, USA) at 25 °C, as previously described [44,45]. Briefly, the fresh Fe^2+^ solution was prepared by fully dissolving FeSO_4_ in MilliQ water. It was then adjusted to pH 3.5 with hydrochloric acid. The protein sample was added to the buffer containing FeSO_4_ (with a final concentration of 128 μM) at a final concentration of 0.5 μM. The absorbance was determined with the addition of ferrozine (with a final concentration of 1 mM). A Fe^2+^ concentration curve was prepared by adding ferrozine into various serially diluted Fe^2+^ concentrations and recording the absorbance of the samples at 560 nm (Appendix A) [46]. The absorbance value of the iron oxidation reaction with the protein sample in relation to time zero (0 s) was considered to be 100%, and the remaining concentrations were transformed into percentages accordingly. BSA was used as the negative control group instead of DzFer.

### 2.8. Microscale Thermophoresis (MST)

MST experiments were conducted using a NanoTemper Monolith NT.115 system (NanoTemper Technologies, Munich, Germany) to determine the equilibrium dissociation constant (K_d_) between DzFer and Ag^+^ or Cu^2+^. The apparatus can allow the analysis of up to 16 solutions leading to the determination of detailed K_d_ in a single run within 30 min. After desalting, DzFer was labeled using a Monolith NT™ Protein Labeling Kit RED-NHS 2nd Generation (NanoTemper Technologies; catalog no. MO-L011) in the supplied labeling buffer according to the manufacturer’s protocol [47]. Then, the labeled proteins were eluted into 50 mM of Tris (pH 7.0) buffer containing 0.05% Tween-20, which was also used as the assay buffer for the MST experiments. A total of 5 μL of labeled protein solution at a concentration of approx. 200 nM was added to the standard treated silicon capillaries (Monolith NT.115 series capillaries; catalog no. MO-K022) with different concentrations of Ag^+^ (from 0.275 nM to 9 µM) or Cu^2+^ (from 0.61 nM to 20 µM) as well as ferrous solution (from 0.61 nM to 20 µM). MST measurements were performed in triplicate at medium MST power in the red channel using 20% excitation power at 25 °C. The data analysis was carried out using the MO Affinity Analysis v2.3 software (NanoTemper Technologies) to fit the curves and calculate the values of K_d_.

### 2.9. Statistical Analysis

The average of multiple measurements (triplicates or more) was shown as the mean ± standard deviation (SD). Comparisons between the two groups were assessed using Student’s *t*-test. In both cases, differences with *p* < 0.05 were considered to be statistically significant. All the data were processed using the OriginPro 9.0 software (OriginLab Corporation, Northampton, MA, USA).

## 3. Results

### 3.1. Characterization of the Molecular Weight and Morphology

The recombinant DzFer containing the SUMO tag was heterologously expressed and purified using a prokaryotic expression system according to the previously reported methods [21]. SDS-PAGE analyses defined a SUMO-tagged DzFer subunit as having a molecular weight (MW) of approx. 30 kDa (Figure 1A). After removing the SUMO tag and further purification, the MW of the AjFER subunit was estimated as approx. 20 kDa (Figure 1B,C). In the Native-PAGE of DzFer, only a single electrophoretic band with an MW of approx. 480 kDa was observed, demonstrating the high purity of DzFer (Figure 1D). To shed light on the effect of the varying pH treatments on the cage stability of DzFer, protein samples (1.0 µM) were exposed to pHs from 2 to 12 (Figure 1D). The Native-PAGE analyses showed that no protein bands of DzFer were observed due to be disassembled under pH 2, whereas most DzFer cages retained the same electrophoretic bands as native ferritin at pHs from 4 to 12. Moreover, Ag^+^-DzFer and Cu^2+^-DzFer proteins were also analyzed using Native-PAGE (Figure 1E). The results showed that these metal-binding proteins exhibited the characteristic bands corresponding to the native DzFer cage, indicating that they were in good agreement with the resulting MW of DzFer. To corroborate this interpretation, all proteins (0.25 mg/mL) (i.e., DzFer at varying pHs from 2 to 12, DzFer, Ag^+^-DzFer, and Cu^2+^-DzFer) were negatively stained with uranyl acetate and observed using TEM (Figure 2). At pH 2, the cage-like structures were hardly seen in the TEM experiments, while the protein molecules exhibited obvious dispersion phenomena at pH 12 (Figure 2A,F). Differently, all proteins presented identical cage-like architectures at pH 4 to 10, demonstrating that the DzFer cage could withstand a broad range of pHs (pH 4–10) (Figure 2B–E). Similarly, most Ag^+^-DzFer and Cu^2+^-DzFer cages maintained spherical shell structures similar to native DzFer cages, suggesting that the binding of Ag^+^ or Cu^2+^ to DzFer did not destroy their protein shell-like structures, although this was likely to have a certain effect on the microenvironment for DzFer (Figure 2G–I).

### 3.2. Characterization of Secondary Structures and Nanoparticle Sizes of Ag^+^ or Cu^2+^ Binding to DzFer

In order to further confirm the above observation, protein hydrodynamic sizes at pH 2–12 were determined using DLS analysis in the solution (Figure 3A). At pH 2, the average DzFer particle diameters were divided into 11.14 ± 0.13 and 234.6 nm, respectively, while the two size populations at pH 12 consisted of a monodispersed distribution of 11.29 ± 0.37 nm and an aggregated macromolecule of 590.0 ± 69.3 nm, respectively. By contrast, the primary size distribution at pHs from 4 to 10 was markedly centered at a single peak around ~13.0 nm in diameter, indicating complete cage formation. As depicted in Appendix A, after the binding of Ag^+^ or Cu^2+^ to the DzFer cage, most Ag^+^-DzFer and Cu^2+^-DzFer cages still exhibited a nearly monodispersed distribution with a hydrodynamic size centered at ~13.0 nm, suggesting that protein aggregation hardly occurred in the solution. The determination of the monodispersed distribution by DLS analyses was then accompanied by CD spectroscopy. It is known that changes in far UV-CD correspond to alterations in the overall secondary structures of proteins. To assess whether pH changes and Ag^+^ or Cu^2+^ affected the secondary structure of ferritins, the CD spectra were monitored according to the changes in ellipticity. As shown in Figure 3B, the DzFer at pHs from 4 to 10 all had very similar two negative ellipticities at 208 and 222 nm in the far-UV spectrum, indicating that protein samples were rich in *α*-helix. The curve-fitting of DzFer (pH 8) with the CDNN software produced the following results: contents of 90.8 ± 0.2% *α*-helix and 9.2 ± 0.2% *β*-turn (Appendix A). In particular, under extreme pH conditions, i.e., 2 and 12, the *α*-helix content in DzFer dramatically decreased, while contents such as *β*-sheet and random coil increased to different extents. Specifically, under the condition of pH 2, the *α*-helix content decreased to 62.37 ± 0.80%, and the *β*-sheet and random coil contents increased to 22.3 ± 2.4% and 4.73 ± 0.50%, respectively. At pH 12, the *α*-helix content drastically decreased to 33.83 ± 2.30%, and the *β*-turn content significantly decreased to 5.0 ± 4.8%, while the *β*-sheet and random coil contents dramatically increased to 53.0 ± 7.3% and 8.13 ± 1.20%, respectively. Most notably, although the CD spectrum of Cu^2+^-DzFer in the 190–240 nm range nearly overlapped with that of DzFer within the acceptable level of the high-tension (HT) voltage (~500 V), a great fluctuation presented in the spectrum of Ag^+^-DzFer was observed within this wavelength range (Appendix A). The results indicated that the binding of Ag^+^ to DzFer much more likely affected the secondary structure of DzFer than that of Cu^2+^. It was evident that the α-helix contents of Ag^+^-DzFer and Cu^2+^-DzFer were decreased by 17% and 4.2%, respectively, and the β-turn contents were increased by 16.9% and 4.1%, respectively, relative to those of DzFer (Appendix A).

### 3.3. Determination of Metal Contents for Ag^+^-DzFer and Cu^2+^-DzFer Cages

As shown in Figure 4A, ICP-MS analyses were carried out to determine the contents of metal ions among these protein samples. The iron content of the purified DzFer was determined to be 56 atoms per cage. In contrast, the binding of Ag^+^ or Cu^2+^ to the DzFer cage resulted in significant decreases in the iron contents, with 14 and 17 atoms per cage, respectively. There were 26 bound silver ions per cage for the Ag^+^-DzFer cage as compared to 113 bound copper ions per cage for the Cu^2+^-DzFer cage. Furthermore, the UV−vis spectra were used to analyze the optical absorption characteristics of these proteins (Figure 4B). Obviously, the absorbance spectra of Ag^+^-DzFer and Cu^2+^-DzFer all had only one maximal protein characteristic absorption peak in the range 260–290 nm, most of which nearly overlapped with that of DzFer. The maximal absorption peaks of them were mainly found at 276 and 277 nm, respectively, except for a 275 nm absorption peak merely observed in Cu^2+^-DzFer. In order to confirm this, the binding affinity related to the interaction of DzFer with Ag^+^ or Cu^2+^ should be investigated. As is well known, MST is a powerful technique for qualitative binding studies to assess molecular interactions based on thermophoresis [48]. MST principle depends on the movement of molecules in a temperature gradient which can be used for studying the binding affinity of target molecules such as proteins, peptides, metal ions, and many others [49,50]. In recent years, MST has developed into one of the newest biophysical methods commercially available to characterize biomolecular affinities of various molecules irrespective of the type or size of the target [51,52]. Therefore, the affinity interaction of DzFer with Ag^+^ or Cu^2+^ was further estimated using MST measurements. As shown in Appendix A, fluorescently labeled DzFer moves from a locally heated region to the outer cold region until a relatively steady state is reached (up to approx. 22 s). The addition of Ag^+^ or Cu^2+^ decreased protein thermophoretic mobility and consequently increased the normalized fluorescence. The MST binding curves of DzFer+Ag^+^ and DzFer+Cu^2+^ revealed that Ag^+^ or Cu^2+^ interact with DzFer at K_d_ values of 3.35 ± 0.65 µM and 8.55 ± 2.85 µM, respectively (Figure 4C,D).

### 3.4. Crystal Structures of Ag^+^-DzFer and Cu^2+^-DzFer Cages

To obtain more direct evidence concerning the binding of Ag^+^ or Cu^2+^ to DzFer, we soaked the DzFer crystals in a silver-soaking solution to identify possible metal-binding sites of Ag^+^ ions in the Ag^+^-DzFer cage (Appendix A). We also prepared single crystals of Cu^2+^-DzFer (Appendix A). This information can provide insights concerning the binding of Ag^+^ or Cu^2+^ to DzFer. The diffraction data from the crystals of Ag^+^-DzFer and Cu^2+^-DzFer were scaled to the maximum resolutions of 1.90 and 2.26 Å, respectively, in the space group I432 (Appendix A). Both Ag^+^-DzFer and Cu^2+^-DzFer structures contained one cage in the asymmetric unit with a visible electron density for residues 1–169 (Figure 5). The overall structures of Ag^+^-DzFer and Cu^2+^-DzFer all exhibited a 24-mer hollow sphere that was built up from identical cages (Figure 5A). The superimposition of cages of DzFer and Ag^+^-DzFer resulted in a root-mean-square deviation (RMSD) value of 0.64 Å over 168 Cα atoms, while the structural superimposition of DzFer with Cu^2+^-DzFer resulted in an RMSD value of 0.74 Å over 168 Cα atoms (Appendix A). On the basis of the X-ray crystallographic analysis, the refined structure of Ag^+^-DzFer showed that a total of 88 Ag^+^ ions were bound to the DzFer nanocage through metal-coordination bonds (Figure 5B). The fold of one subunit in Ag^+^-DzFer was composed of an antiparallel four-helix bundle together with a tiny helix E located at the end of the DE loop, where five Ag^+^-binding sites were observed near the helix A, the BC loop, and the CD loop (Figure 5C). Notably, the ferroxidase site was assigned one Ag^+^ ion because putting the Fe^3+^ here could produce a prominent negative *Fo*–*Fc* electron density map. (Although the refined structure is modeled as Fe^3+^, the oxidation state of iron ion is uncertain, and from here on out, it is referred to as Fe.) Similarly, the refined Cu^2+^-DzFer structure revealed that there were 56 Cu^2+^ ions bound to the DzFer nanocage by metal-coordination bonds (Figure 5D). The five-helix fold stemming from one Cu^2+^-DzFer subunit accommodated five Cu^2+^-binding sites, which were mainly located near the helix D and CD loop (Figure 5E). Although we also found some positive electron densities surrounding the ferroxidase center, it was not clear whether the electron density was occupied by Cu^2+^ ions. When the Cu^2+^-DzFer model was refined by positioning Fe^3+^ on the di-iron ferroxidase sites, the positive *Fo*–*Fc* electron density maps were observed (Figure 5E).

In the Ag^+^ cocrystal structure of DzFer, two Ag^+^-binding sites (Ag1 and Ag2) were observed inside the three-fold channel, whereas the other three Ag^+^-binding sites (Ag3, Ag4, and Ag5) were observed at the outer surface groove of the Ag^+^-DzFer shell through metal-coordination bonds (Figure 6A). Ag1 was ligated by the carboxylate groups of Glu130 residues at distances of 2.7 Å and three water molecules at distances of 2.3 Å. Ag2 was coordinated with the carboxylate groups of the Asp127 (at distances of 3.2 Å) and Glu130 (at distances of 2.9 Å) residues and three water molecules (at distances of 3.3 Å) (Figure 6B). Furthermore, Ag1 and Ag2 were observed with a Ag-Ag distance of 2.5 Å. Ag3 was found at the ferroxidase center in the Ag^+^-DzFer cage, and it was bound to the carboxylate groups of the Glu58 and Glu103 residues with distances of 3.7 and 3.3 Å, respectively. Two water molecules were also observed (at distances of 2.8 and 2.9 Å) (Figure 6C). The binding site of Ag4 was located on the N-terminal of the A-helix and BC loop near the three-fold channel in the Ag^+^-DzFer cage, where the Cys12 and Asp122 residues were involved in coordination with Ag^+^ ions, as shown in Figure 6D. Ag4 was ligated by the thiolate group of the Cys12 residue at a distance of 3.1 Å, the carboxylate group of the Asp122 residue at a distance of 2.7 Å, and one water molecule at a distance of 2.7 Å. At the site of Ag5, the Ag^+^ ion was coordinated with the thioether group of Met31 at a distance of 3.2 Å, the hydroxyl group of Ser35 at a distance of 3.8 Å, and one water molecule at a distance of 2.7 Å (Figure 6E).

In the Cu^2+^-DzFer structure, five Cu^2+^ ions were observed bound at the three-fold channel and its surface groove. Among these, four were close to each other, and two coordinated with the carboxylate groups of the Asp and Glu residues inside the three-fold pore (Figure 7A). Specifically, one Cu^2+^ ion (Cu1) was located near the exterior opening of the three-fold channel, and it was coordinated by six water molecules alone (at distances of 3.6 and 3.7 Å, respectively), accompanied by the Cu1-Cu2 intermetallic distance of 3.6 Å (Figure 7B). Moreover, six carboxylate groups of the Asp127/Glu130 residues stemming from three symmetric subunits point toward the center inside the three-fold channel pore. They coordinated with two Cu^2+^ ions (Cu2 and Cu3) at the Cu2-Cu3 intermetallic distance of 2.6 Å. Among these, Cu2 was bound to the carboxylate groups of the Glu130 residues at distances of 2.9 and 3.6 ± 0.1 Å, respectively, and three water molecules at distances of 3.5 Å, while Cu3 was ligated by the carboxylate groups of the Glu130 (at distances of 2.9 Å) and Asp127 (at distances of 3.4 Å) residues and three water molecules at distances of 3.6 Å. In addition, there was also another Cu^2+^ ion (Cu4) coordinated by three water molecules alone, at distances of 3.2 Å, near the interior exits of the three-fold symmetry pore, at the Cu3-Cu4 distance of 4.9 Å (Figure 7B). Moreover, the other Cu^2+^ ion (Cu5) was located at the outer surface groove of the Cu^2+^-DzFer cage, and it was coordinated with the thiolate group of the Cys12 residue (at a distance of 3.2 Å), the carboxylate group of the Asp122 residue (at a distance of 2.7 Å), and one water molecule (at a distance of 2.6 Å), as illustrated in Figure 7C.

### 3.5. Effect of Binding Ag^+^ or Cu^2+^ on the Ferroxidation Reaction by DzFer

As seen from the above results obtained from the CD spectra, although the binding of Ag^+^ or Cu^2+^ to DzFer caused some obvious changes as compared to those of DzFer, their crystal structures did not seem to be significantly affected based on the structural superposition (Appendix A). To further investigate the effect of Ag^+^-DzFer and Cu^2+^-DzFer on the secondary structures after Fe^2+^ uptake, the contents of *α*-helix, *β*-turn, *β*-sheet and random coil were calculated based on the CD spectra within the acceptable level of the HT voltage (~500 V) as depicted in Figure 8A and Appendix A. The results indicated that the *α*-helix content during the uptake of Fe^2+^ by DzFer (DzFer+Fe^2+^) was 84.5 ± 0.7%, while the *β*-sheet content was 15.5 ± 0.7% (Appendix A). After Fe^2+^ uptake by Ag^+^-DzFer (Ag^+^-DzFer+Fe^2+^), the *α*-helix content decreased down to 72.2 ± 0.7%, and the *β*-turn content increased to 15.5 ± 0.7%. Correspondingly, the *α*-helix and *β*-turn contents during the uptake of Fe^2+^ by Cu^2+^-DzFer (Cu^2+^-DzFer+Fe^2+^) changed to 85.7 ± 0.4% and 14.3 ± 0.4%, respectively. Subsequently, the iron oxidation activities catalyzed by DzFer, Ag^+^-DzFer and Cu^2+^-DzFer were evaluated by incubating the protein samples in the presence of Fe^2+^ and determining the conversion to Fe^3+^. This was compared using BSA as the control (Figure 8B and Appendix A). Although the DzFer, Ag^+^-DzFer, and Cu^2+^-DzFer presented consistently increasing trends when oxidizing Fe^2+^, their ferroxidase activities displayed some significant differences on the basis of the oxidation activity progress curves (Appendix A). Specifically, the ferroxidation reaction of DzFer exhibited a higher oxidation activity than that of Ag^+^-DzFer and Cu^2+^-DzFer, whereas the ferroxidase activity of Cu^2+^-DzFer significantly increased as compared to that of Ag^+^-DzFer. As shown in Figure 8C, the protein samples were able to rapidly oxidize the available Fe^2+^ into Fe^3+^. After 120 s of incubation, both DzFer and Cu^2+^-DzFer oxidized approx. 83% of the available Fe^2+^, whereas Ag^+^-DzFer oxidized approx. 67% of the available Fe^2+^ (Appendix A).

Taking the above results together, it was concluded that the ferroxidase activities of Ag^+^-DzFer and Cu^2+^-DzFer were severely affected as compared with native DzFer, which was mostly attributed to Ag^+^ or Cu^2+^ binding in the DzFer cage disturbing the interaction of these polymers with Fe^2+^ ions. To verify the hypothesis related to these interactions, MST analyses were performed to assess the binding affinities between Fe^2+^ and DzFer, Ag^+^-DzFer, and Cu^2+^-DzFer. The MST time traces of all scanned solutions showed a successive decline in signal, indicating that the thermophoresis of protein samples presented a concentration-dependent change, which clearly indicated the binding of Fe^2+^ to DzFer, Ag^+^-DzFer, and Cu^2+^-DzFer (Appendix A). The thermophoretic traces demonstrate striking differences between the high and low concentrations in response to thermophoresis. Additionally, the binding curves displayed typical decreasing or increasing sigmoidal shapes (Figure 8D–F). The MST results revealed significant binding interactions between the Fe^2+^ ion and DzFer, Ag^+^-DzFer, and Cu^2+^-DzFer with different K_d_ values. In detail, Fe^2+^ was able to strongly bind to DzFer with high efficiency, resulting in the lowest K_d_ value of 2.12 ± 0.88 µM (Figure 8D). Both Ag^+^-DzFer and Cu^2+^-DzFer were observed to bind with relatively high affinities to Fe^2+^ as expected, with K_d_ values of 18.0 ± 11.5 µM and 7.58 ± 4.16 µM, respectively (Figure 8E, F).

## 4. Discussion

It is generally accepted that ferritin is strongly correlated with a ubiquitous iron storage protein. Moreover, the uptake of metal ions in ferritin is not limited to Fe^2+^, and the homopolymer has a strong negative inner electrostatic potential and multiple ion chelation centers, allowing for specific interactions with various cations (e.g., Ag^+^ and Cu^2+^) [17,53]. In this study, a marine invertebrate ferritin named DzFer was identified from *D. zhejiangensis*, which resided at the bottom of muddy shrimp ponds [21]. In this environment, DzFer very probably comes into contact with heavy metal ions such as Ag^+^ and Cu^2+^. To elucidate the effect of Ag^+^ or Cu^2+^ on the relationship between the structure and function of DzFer, we characterized the structural and biochemical properties of Ag^+^-DzFer and Cu^2+^-DzFer.

Despite their rigid architecture, ferritin molecules can be dissociated or denatured under extreme conditions (e.g., strong acids/alkali, heavy metal ions), and thus, their robust cage-like structures are likely to be destroyed, affecting their functions [22]. Thus, the effect of pH and Ag^+^ or Cu^2+^ association on the DzFer cage was assessed via biochemical and structural experiments in the pH range of 2 to 12 and under treatment with Ag^+^ or Cu^2+^. Recombinant DzFer displayed an MW of approx. 20 kDa for a single subunit, as demonstrated using SDS-PAGE, forming a 24-mer hollow multipolymer structure with an estimated MW of more than 440 kDa (Figure 1C,D), which is very similar to classical ferritins from other marine invertebrates [54]. As previously reported for the recombinant horse L-chain apoferritin, which is highly stable in response to pH changes [22], the 24-mer cage of DzFer also exhibited adaptative stability at varying pH conditions (pH 4–10), as evidenced by the Native-PAGE and TEM results (Figure 1D and Figure 2B–E). Under highly acidic conditions (pH > 2), the DzFer cages became extremely unstable, as suggested by the Native-PAGE results. They even became dissociated into individual subunits, as observed in the TEM image (Figure 2A), which is consistent with the results of human H ferritin (HuHF) [55] and other marine invertebrate ferritins [56]. In contrast, despite partially collapsing and dissociating (i.e., disassembling) at pH 12, the majority of DzFer cages apparently retained their shapes, as revealed in the TEM images (Figure 2F). Combining the results from the hydrodynamic diameter estimated by DLS and the secondary structures determined by CD spectra (Figure 3A,B), it can be inferred that DzFer has biological characteristics similar to those of mammalian ferritins, for example, an ability to retain intact spherical ferritin structures over a wide pH range, e.g., between 2 and 12.

As is well-known, with the exception of iron, there are a number of metal ions that can bind to the cage-like shell structure of ferritin, including DzFer [12,20]. On these grounds, we attempt to establish whether the binding of Ag^+^ or Cu^2+^ to DzFer has an effect on the functional features of DzFer associated with the iron permeability and then biomineralization into the interior surface of the protein cage. To this end, Native-PAGE analyses of Ag^+^-DzFer and Cu^2+^-DzFer were performed to explore their aggregation states in solution, revealing almost the same electrophoretic behavior as native DzFer in nondenaturing gel electrophoresis (Figure 1E), suggesting that they were purified to homogeneity. This observation was further demonstrated by the TEM results (Figure 2G–I), indicating that the typical shell-like structures of these nanocages were consistent with the structure of native ferritin. Moreover, they were shown to be uniform and monodispersed phenomena in diameter, as evidenced by the DLS results (Appendix A), suggesting the maintenance of a certain amount of their physiological activity [10]. To further corroborate this interpretation, CD spectroscopy was then applied to obtain more information concerning the changes in the secondary structures of DzFer, Ag^+^-DzFer, and Cu^2+^-DzFer (Appendix A). As compared with DzFer, we observed a significant decrease in the *α*-helix content in Ag^+^-DzFer (approx. 17%) and Cu^2+^-DzFer (approx. 4.2%) (Appendix A). Nonetheless, whether in DzFer or in metal-bound DzFer, the contents of secondary structures (i.e., α-helix) remained relatively stable based on the calculation by the STRIDE server [41], quite distinct from the results of CD analyses (Appendix A). Notably, the contents of a random coil in Ag^+^-DzFer and Cu^2+^-DzFer were increased by 2.3% and 1.1%, respectively, in comparison with DzFer. This was probably attributed to the interaction of DzFer with Ag^+^ or Cu^2+^ indirectly affecting the formation of hydrogen bonds and resulting in structural changes in ferritin [57]. In addition, we further confirmed the contents of 26.01 ± 0.12 Ag atoms per cage in Ag^+^-DzFer and 113.26 ± 0.37 Cu atoms per cage in Cu^2+^-DzFer via ICP-MS analysis (Figure 4A), suggesting that both Ag^+^ and Cu^2+^ were able to bind to the DzFer nanocage through metal coordination bonding. Although it is reported that the Ag^+^ or Cu^2+^ can be bound to the ferritin nanocage via the interactions with the Cys and/or His residues [58,59], the results that the UV−vis spectra of both Ag^+^-DzFer and Cu^2+^-DzFer almost overlap with that of DzFer are not sufficient to illustrate the existence of such interactions (Figure 4B). In order to further investigate these interactions, the binding affinity of the interactions of DzFer with Ag^+^ or Cu^2+^ was evaluated using MST experiments (Figure 4C,D). We found that both Ag^+^ and Cu^2+^ had high affinities for DzFer, with K_d_ values of 3.35 ± 0.65 and 8.35 ± 2.85 μM, respectively. Obviously, MST results revealed that there were some significant interactions of DzFer with Ag^+^ or Cu^2+^, whereas the binding reaction for DzFer with Ag^+^ was much stronger than that of Cu^2+^. However, it is still insufficient for MST analysis to measure the binding affinity under equilibrium conditions, such as isothermal titration calorimetry (ITC), and identify the specific metal-binding sites. Nonetheless, the binding reaction is definitely not a nonspecific electron static interaction because this point can be best demonstrated by the results of Cu^2+^/Ag^+^ coordination in the crystal structures of Ag^+^-DzFer and Cu^2+^-DzFer. Together, these results show that both Ag^+^ and Cu^2+^ are capable of binding in DzFer; however, there is still a lack of structural evidence to confirm this.

To establish the detailed structural features in order to profile the metal-binding sites, we determined the three-dimensional crystal structures of Ag^+^-DzFer and Cu^2+^-DzFer using an X-ray crystallographic analysis (Figure 5A). Despite showing that Ag^+^ or Cu^2+^ bound to DzFer, the superimposition analysis revealed that the overall architectures of Ag^+^-DzFer and Cu^2+^-DzFer were almost identical and were highly consistent with the crystal structure of native DzFer [21]. In this study, our structural data, coupled with the biochemical and ICP-MS analyses, suggested that there were five Ag^+^ binding sites per subunit for a total of 88 Ag^+^ ions per Ag^+^-DzFer cage (Figure 5B,C) and five Cu^2+^ binding sites per subunit for a grand total of 56 Cu^2+^ ions per Cu^2+^-DzFer cage (Figure 5D,E). To date, there is a widespread belief that the three-fold channels in ferritins are potential gates for the entry of metal ions into the cage [12]. A previous X-ray crystal structure analysis also revealed that the three-fold channels of DzFer were surrounded by regions of predominantly negative potential, allowing the passage of metal ions into its interior cavity [21]. This demonstrates that the three-fold channel is the major entrance into the ferritin cage [21]. In the Ag^+^-DzFer structure, although few studies regarding ferritin with structural data to support this, we observed an argentophilic interaction between Ag1 and Ag2 (the Ag-Ag distance of 2.5 Å) [60], and they coordinated with the negatively charged residues Asp127 and Glu130 at the three-fold channel (Figure 6A,B). Most notably, one Ag^+^ ion (Ag3) was designated as binding at the ferroxidase center based on the strong electron density maps around the ferroxidase site, which was most likely attributed to the strong binding ability of Ag^+^ ions with the DzFer cage (Figure 6C). Given that silver ions have highly thiophilic features, we observed that there was a fourth Ag^+^-binding site (Ag4) located on the N-terminal of the A-helix at the three-fold channel of DzFer, where the Cys12 and Asp122 residues were involved in coordination with Ag^+^ (Figure 6D). Similar coordination of Ag^+^ to the Cys and Asp residues was also observed in a cyanobacterial metallochaperone CopM structure [15]. As another sulfur-containing amino acid, it has been suggested that methionine plays a crucial role in efficient silver incorporation in PfFt [18]. On this basis, we also observed a fifth Ag^+^-binding site (Ag5) buried near the interior of the three-fold pore in the Ag^+^-DzFer cage, where Ag^+^ was coordinated by the side chains of the two Met31 and Ser35 residues (Figure 6E). In the Cu^2+^-DzFer structure, four Cu^2+^-binding sites were assigned based on the clear spherical densities around the three-fold pore in the electron density maps (Figure 7A). Two Cu^2+^-binding sites (Cu1 and Cu4) were observed near the exterior and interior openings of the three-fold channel, and both were entirely coordinated by water molecules alone (Figure 7B). Of note, the variable width of the three-fold channel contained the narrowest dimension, i.e., 5.4 Å, formed by carboxylate side chains of three conserved Glu130 residues. This is mostly consistent with the three-fold channel in FrMF [30]. The architecture likely results in the metal ions inside the three-fold pore having less access to water molecules and thus directly interacting with the protein ligands [61]. Considering this, the other two Cu^2+^-binding sites (Cu2 and Cu3) were observed inside the center of the three-fold channel, where they were coordinated with the carboxylates of conserved Asp127 and Glu130 residues and water molecules (Figure 7B). This is very similar to the coordination environment of Mg^2+^ in FrMF [30] and Cu^2+^ in *Tegillarca granosa* ferritin (TgFer) [10]. Similar to the coordination environment with the Ag4 site, the fifth Cu^2+^-binding site (Cu5) was shown to be mainly composed of conserved Cys12 and Asp122 residues (Figure 7C). Taken together, these results suggest that both Ag^+^ and Cu^2+^ are able to bind to the DzFer cage via metal-coordination bonds, and their binding sites are mainly located at the three-fold channel of DzFer. In addition, Ag^+^ appeared to bind preferentially at the ferroxidase site of DzFer and had a higher selectivity for sulfur-containing amino acid residues as compared with Cu^2+^.

Numerous studies show that many different types of metal ions, other than iron ions, are capable of binding to ferritins, which can potentially affect the rate of catalysis of Fe^2+^ oxidation in the ferroxidase center [10,12]. For this reason, it was necessary in this study to supplement the monitoring of the oxidation reaction rates of Fe^2+^ ions catalyzed by Ag^+^-DzFer and Cu^2+^-DzFer. After the uptake of Fe^2+^ ions (Figure 8A), the stability of the DzFer+Fe^2+^, Ag^+^-DzFer+Fe^2+^, and Cu^2+^-DzFer+Fe^2+^ cages may be significantly affected in accordance with the obviously reduced *α*-helix contents as compared with native DzFer (Appendix A). It is worth noting that the ferroxidase activity of Ag^+^-DzFer and Cu^2+^-DzFer is significantly lower than that of DzFer, and the iron oxidation rate of Ag^+^-DzFer is much lower than that of Cu^2+^-DzFer (Figure 8B,C). This may be attributable to Ag^+^ binding at the ferroxidase site. Furthermore, the MST results further demonstrated that native DzFer had a much higher binding affinity for Fe^2+^ than Ag^+^-DzFer and Cu^2+^-DzFer. This was in contrast with Cu^2+^-DzFer, which exhibited more affinity to Fe^2+^ than of Ag^+^-DzFer (Figure 8D–F), implying that the binding of Ag^+^ to DzFer could have a certain inhibitory effect on the interaction of Fe^2+^ with DzFer. These results lead us to believe that the binding of Ag^+^ or Cu^2+^ to DzFer largely impedes the entrance of Fe^2+^ into the protein shell through the three-fold channels, affecting Fe^2+^ interactions with Ag^+^-DzFer and Cu^2+^-DzFer, and inhibiting their fast oxidation at the ferroxidase center.

## 5. Conclusions

In this work, we prepared recombinant DzFer obtained from a marine invertebrate worm and determined the structural and biochemical properties of Ag^+^-bound DzFer (Ag^+^-DzFer) and Cu^2+^-bound DzFer (Cu^2+^-DzFer). Our results demonstrate that DzFer is able to withstand certain extreme pH fluctuations, similar to other mammalian ferritins, and has diverse capabilities as regards interactions with Ag^+^ or Cu^2+^ ions. An X-ray crystallographic analysis revealed that both Ag^+^ and Cu^2+^ were able to bind to the DzFer cage through metal-coordination bonds, and these metal-binding sites were mainly located inside the three-fold channel of DzFer. Moreover, Ag^+^ appeared to bind preferentially to the ferroxidase site of DzFer and had a higher selectivity for sulfur-containing amino acid residues as compared with Cu^2+^. Nonetheless, the binding of Ag^+^ or Cu^2+^ to DzFer, especially Ag^+^, very likely inhibits iron’s rapid oxidation by preventing Fe^2+^ from entering the protein shell through the three-fold channel or even binding at the ferroxidase site. This warrants future investigations into the Ag^+^/Cu^2+^-binding sites in DzFer via site-directed mutagenesis and the crucial role of Ag^+^ in inhibiting the ferroxidase activity of DzFer.

## Figures and Tables

**Figure 1 polymers-15-01297-f001:**
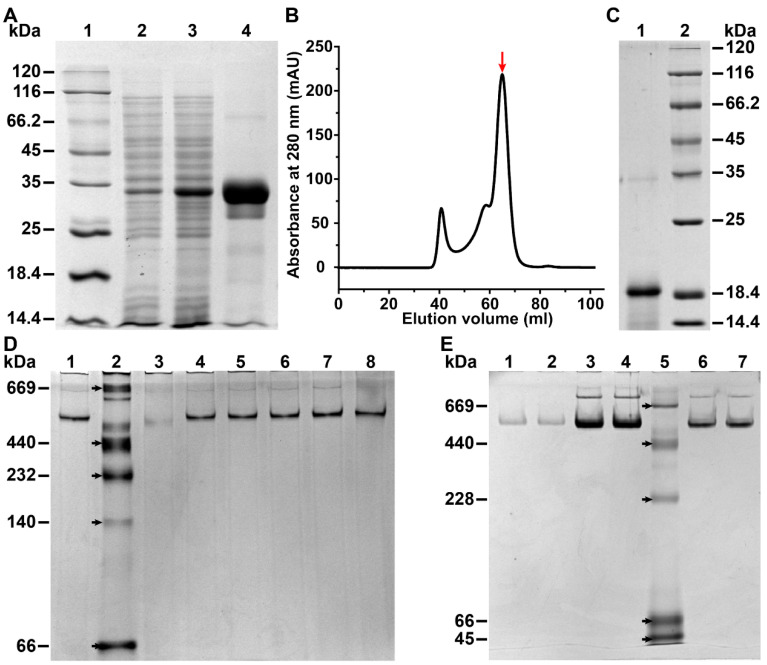
(**A**) Analysis of recombinant DzFer using SDS-PAGE. Lane 1: middle molecular weight markers; lane 2: negative control without induction of DzFer; lane 3: induced expression of DzFer with final IPTG concentration of 0.5 mM for 18 h at 18 °C; lane 4: purified recombinant DzFer with the SUMO tag. (**B**) Superdex 200 gel-filtration chromatography profile of DzFer, where the target protein is shown by a red arrow. (**C**) SDS–PAGE profile showing an approximate molecular weight of 20 kDa. Lane 1: purified DzFer without the SUMO tag; lane 2: middle molecular weight markers. (**D**) Native-PAGE analysis of DzFer at varying pH conditions (pH 2–12). Lane 1: DzFer; lane 2: protein markers; lanes 3–8: pH conditions of DzFer in the following order: pH 2, pH 4, pH 6, pH 8, pH 10, and pH 12. (**E**) Native-PAGE analysis of DzFer, Ag^+^-DzFer, and Cu^2+^-DzFer proteins. Lanes 1–2: DzFer; lanes 3–4: Ag^+^-DzFer protein; lane 5: protein markers; lanes 6–7: Cu^2+^-DzFer protein.

**Figure 2 polymers-15-01297-f002:**
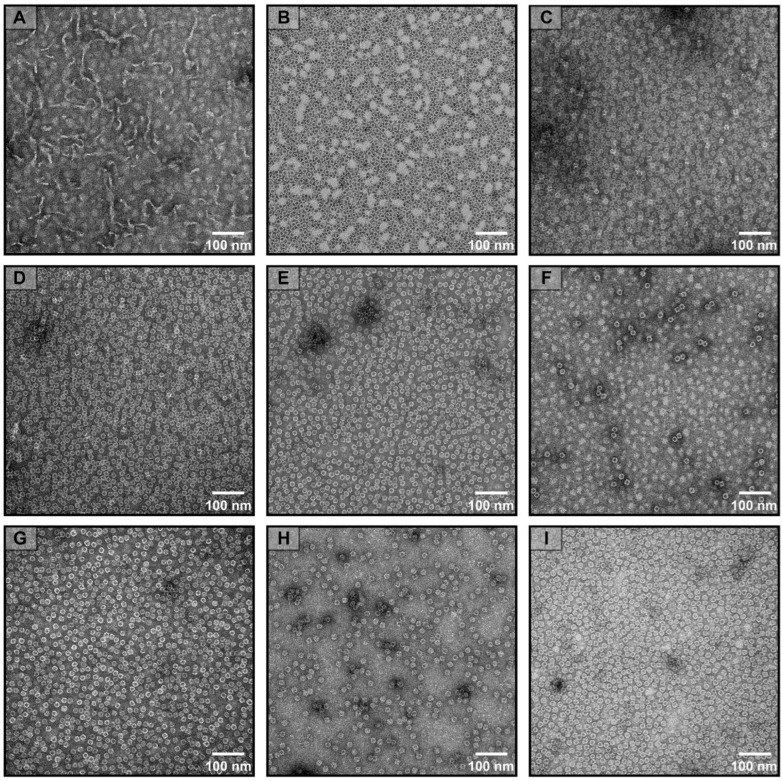
Transmission electron microscope scanning images corresponding to different pH conditions of DzFer and metal-bound DzFer. TEM images under pH 2 (**A**), pH 4 (**B**), pH 6 (**C**), pH 8 (**D**), pH 10 (**E**), and pH 12 (**F**) conditions. (**G**–**I**) TEM images of DzFer, Ag^+^-DzFer, and Cu^2+^-DzFer proteins.

**Figure 3 polymers-15-01297-f003:**
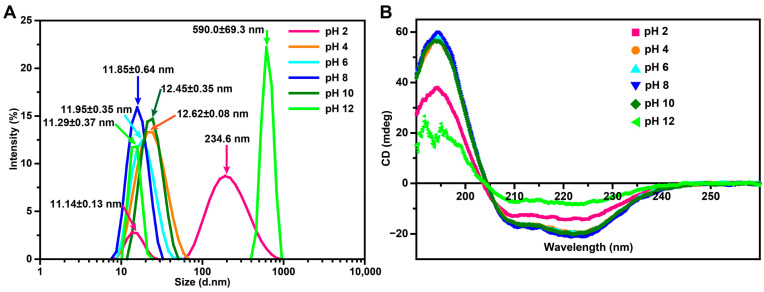
(**A**) DLS analysis of DzFer at pHs 2–12. (**B**) Effects of pH treatment (pH 2–12) on the secondary structures of DzFer.

**Figure 4 polymers-15-01297-f004:**
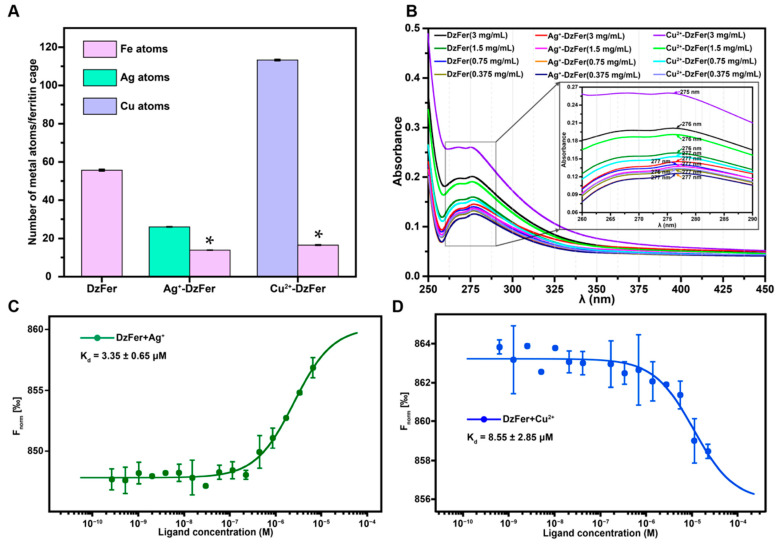
(**A**) Evaluation of the amount of Fe, Ag, and Cu atoms bound to the protein cage by ICP-MS. The protein concentrations were determined using a BCA kit. Data are represented as the mean ± standard error of the mean (SEM). The significant differences are indicated by * (*p* < 0.05) as compared to the DzFer group. (**B**) UV–vis absorption spectra of DzFer, Ag^+^-DzFer, and Cu^2+^-DzFer proteins at the concentrations of 3, 1.5, 0.75, and 0.375 mg/mL, respectively. The inset (right) highlights the comparison of protein characteristic absorption peaks between 260 and 290 nm. Analysis of the interaction between DzFer and Ag^+^ (**C**) or Cu^2+^ (**D**) ions via microscale thermophoresis (MST). Error bars represent the range of data points obtained from three independent measurements.

**Figure 5 polymers-15-01297-f005:**
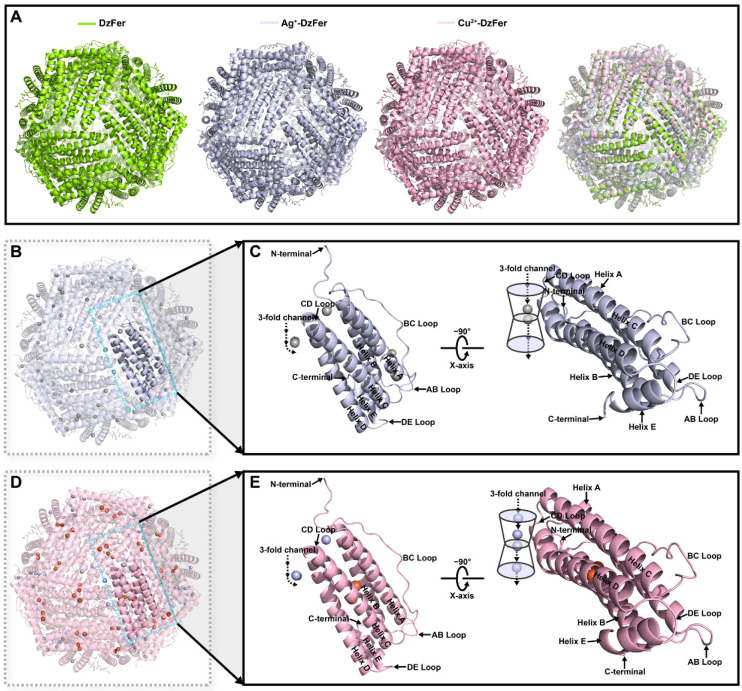
Cartoon representations of DzFer, Ag^+^-DzFer, and Cu^2+^-DzFer along the three-fold symmetry axes based on the X-ray crystallographic analysis. (**A**) Superimposed structures of DzFer (light-green cage; PDB ID, 7EMK), Ag^+^-DzFer (blue-white cage), and Cu^2+^-DzFer (light-pink cage) in cartoon representation. (**B**) Overall architecture of Ag^+^-DzFer. (**C**) One subunit, stemmed from 24-mer cage-like structure of Ag^+^-DzFer, is shown as a detailed secondary structure cartoon and colored blue-white. (**D**) Overall structure of Cu^2+^-DzFer. (**E**) One subunit, stemmed from 24-mer cage-like structure of Cu^2+^-DzFer, is presented as a detailed secondary structure cartoon and colored light pink. Transparent cartoon view with α-helices shown as cartoons and bound metal ions as spheres. Bound metal ions are shown as orange spheres for Fe^3+^ ions, silver-grey spheres for Ag^+^ ions, and light-blue spheres for Cu^2+^ ions.

**Figure 6 polymers-15-01297-f006:**
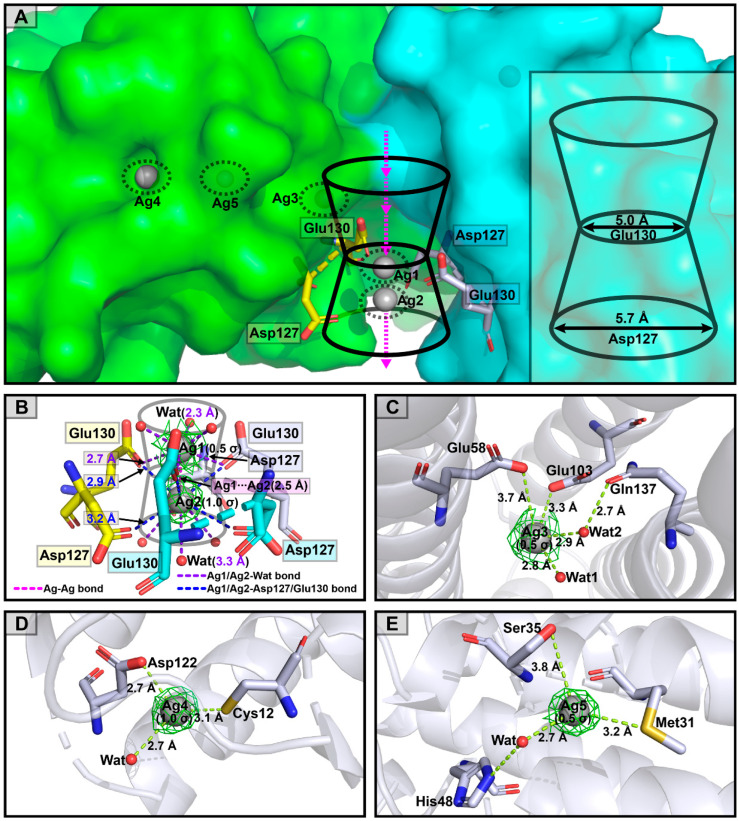
Silver ions traffic into the DzFer cage from the external environment. Protein residues from different subunits are indicated as sticks with yellow, cyan or blue-white carbons. Silver ions are represented as silver-grey spheres. (**A**) Stereo diagrams of the three-fold channel viewed from the outer surface of the Ag^+^-DzFer cage (side view). (**B**) Two Ag^+^ ions (Ag1 and Ag2) are coordinated with Asp127 and Glu130 residues (yellow, cyan, or blue-white sticks) inside the three-fold channel of the Ag^+^-DzFer cage. Silver ion itself and its coordination with an amino acid residue and a water molecule is shown as carmine, blue, and purple dashed lines. (**C**) One Ag^+^ ion (Ag3) can be observed coordinated at the ferroxidase center. (**D**) Silver coordination site on the outer surface of Ag^+^-DzFer cage, where one Ag^+^ ion (Ag4) is coordinated by Cys12 and Asp122 residues from one subunit. (**E**) Another coordination of Ag^+^ ion (Ag5) by Met31 and Ser35 residues can be observed inside one subunit. The electron densities for silver ions (green meshes) are contoured at 0.5 σ (Ag1, Ag3, and Ag5) and 1.0 σ (Ag2 and Ag4).

**Figure 7 polymers-15-01297-f007:**
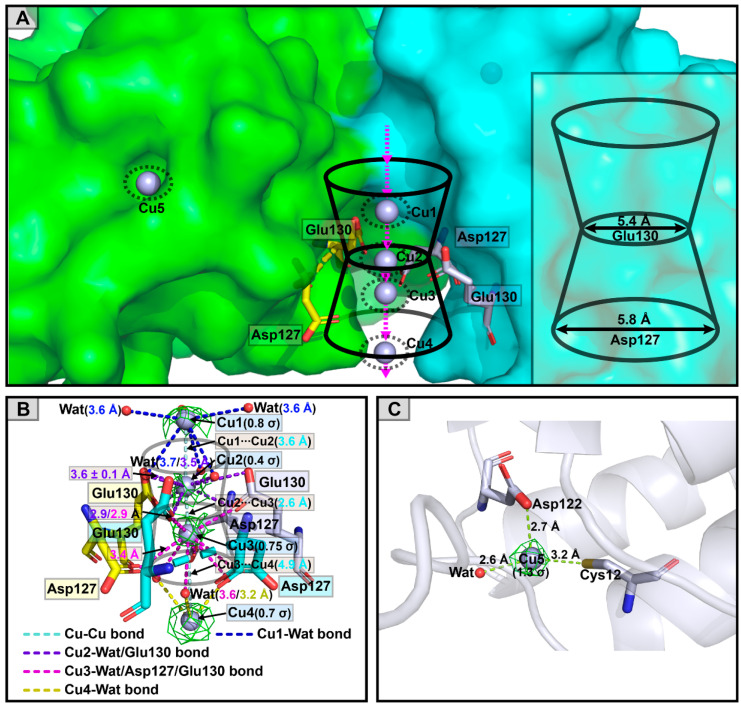
Coordination structures of copper ion binding sites at the Cu^2+^-DzFer cage. Representative amino acid residues from different subunits are shown as yellow, cyan, or blue-white sticks. Copper ions are represented as light-blue spheres. (**A**) Stereo view from the outer surface of the three-fold channel inside the Cu^2+^-DzFer cage (side view). (**B**) Four copper ions (Cu1, Cu2, Cu3, and Cu4) are located inside the three-fold channel of Cu^2+^-DzFer, and they are ligated by Asp127 and Glu130 residues and water molecules. The copper ion itself and its coordination with an amino acid residue and a water molecule is indicated as pale cyan, blue, purple, carmine, or yellow dashed lines. (**C**) Copper coordination (Cu5) on the outer surface of Cu^2+^-DzFer, with one copper ion coordinated by Cys12 and Asp122 residues from one subunit. The resulting 2*Fo–Fc* electron density maps of copper ions (green meshes) are contoured at 0.8 σ (Cu1), 0.4 σ (Cu2), 0.75 σ (Cu3), 0.7 σ (Cu4), and 1.3 σ (Cu5).

**Figure 8 polymers-15-01297-f008:**
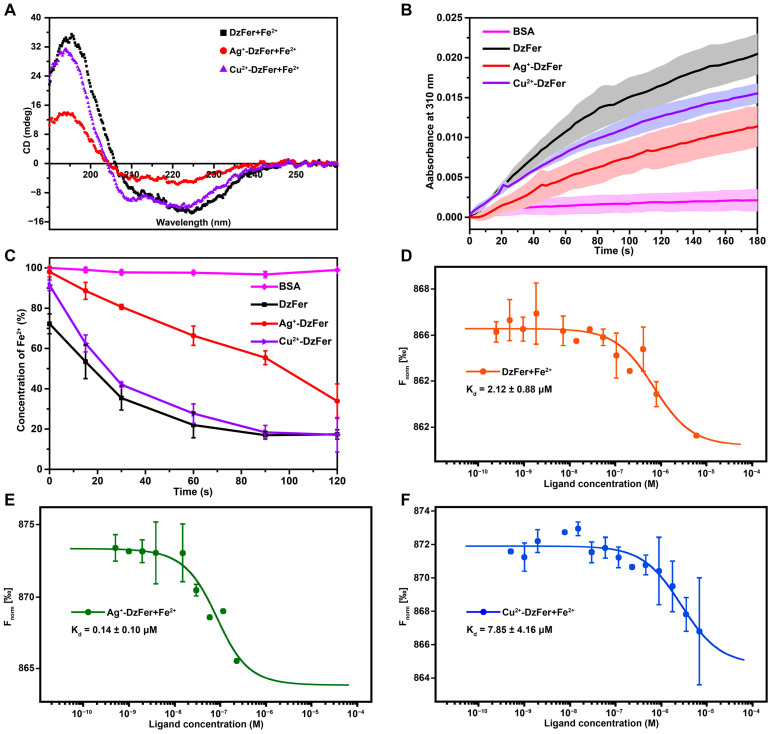
(**A**) Far-UV CD spectra underlying the secondary structure changes in DzFer, Ag^+^-DzFer, and Cu^2+^-DzFer after Fe^2+^ ion uptake. (**B**) The progress curves of the oxidation of Fe^2+^ into Fe^3+^ ion related by BSA, DzFer, Ag^+^-DzFer, and Cu^2+^-DzFer were monitored at 310 nm at room temperature. Solid lines indicate the average of three technical replicates, and shaded areas represent the standard deviation (SD) from the mean. (**C**) The concentration of Fe^2+^ was detected based on monitoring the absorbance of the ferrozine–Fe^2+^ complexes at 560 nm. Experiments were performed independently in triplicate (*n* = 3), and the data were reported as mean ± SD. Representative MST fitting curves of Fe^2+^ ion binding to DzFer (**D**), Ag^+^-DzFer (**E**), and Cu^2+^-DzFer (**F**) proteins. Error bars represent the range of data points obtained from three independent measurements.

## Data Availability

The atomic coordinates and structural factors of Ag^+^-DzFer and Cu^2+^-DzFer have been deposited in the PDB database (http://www.wwpdb.org/) under the accession codes 8GY1 and 8HCT.

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
