# Peer review of "Structural and Biochemical Characterization of Silver/Copper Binding by Dendrorhynchus zhejiangensis Ferritin"

_polymers, 2023, doi:10.3390/polym15051297_

Round 1

Reviewer 1 Report

see pdf file

Author Response

Response to Reviewer #1 Comments (highlight in yellow)

  1. P-1, l-38: “In 1937, it was demonstrated that ferritin is a class of well-conserved iron storage and detoxification proteins commonly found in the cytosol of both prokaryotes and eukaryotes”. This sentence suggests that the authors are citing a paper that they have not read (ref [3]); the paper describes the crystallization of a single species only, namely horse spleen ferritin. There is no mention of prokaryotes, and there is no allusion to occurrence in eukaryotes other than horse. Please, correct the text.

Reply: Thank you for your suggestions. The reference “Laufberger, V. Sur la cristallisation de la ferritine. Soc Chim Biol 1937, 19, 1575–1582” had been replaced by “Bou-Abdallah, F. The iron redox and hydrolysis chemistry of the ferritins. Biochim Biophys Acta Gen Subj 2010, 1800, 719–731” in page 24, line 785-786 in the revised manuscript.

  1. p-2, l-51: “.. accommodating up to ~4500 iron atoms”. This number has recently been shown to be a long-standing error in the literature. The actual maximum loading capacity of ferritins is circa 2000-3000 Fe, see Metallomics (2022) 14 : https://doi.org/10.1093/mtomcs/mfac063. Please correct the number.

Reply: Thank you for your suggestions. The related contents had been corrected in page 2, line 51. Moreover,  The reference “Plays, M.; Müller, S.; Rodriguez, R. Chemistry and biology of ferritin. Metallomics 2021, 13, mfab021” had been replaced by “Hagen, W. R. Maximum iron loading of ferritin: half a century of sustained citation distortion. Metallomics 2022, 14, mfac063” in page 25, line 796-797 in the revised manuscript.

  1. p-2, l-92: “Inspired by the living conditions of Dh, which usually inhabits heavy metal-rich environments...”. (1) is there any information in the literature about this heavy metal-rich environment, or is this a guess of the authors. Please specify in the text. (2) Is there any experimental indication that Cu(II) is freely available in the cytosol and not bound to copper transport protein(s) directed towards biosynthesis of natural copper proteins? And is there any experimental proof that Ag(I) actually enters the cells of Dh? Please specify in the text.

Reply: Thank you for your suggestions. As previously reported, Dendrorhynchus zhejiangensis are one of the marine invertebrates belonged to Neatinea, Heteronemertea, Lineidae, Dendrorhynchus, and they live in the bottom of shrimp ponds, where the concentration of heavy metal is usually higher than that in seawater [1]. However, we are sorry for the thoughtless consideration of these research contents, so it is not clear whether Cu(II) is freely available in the cytosol and it can interact with the copper transport protein in D. zhejiangensis. Moreover, there is also still a lack of conclusive evidence on whether Ag(I) can enter the cells of D. zhejiangensis. The related reference had been added in page 25, line 820-821 in the revised manuscript.

  1. p-3, l-130: “.. with a pH value gradient from 2 to 12”. The authors do not use a gradient, but a set of discrete pH values in the range 2-12. Also, the pK’s of phosphate are 2.1, 7.2, and 12.4, which is where the buffer capacity is maximal. At intermediate values (4, 10) the capacity is orders of magnitude less. Did the authors check the final pH of the preparations of protein in phosphate buffer? And what was the buffer concentration?

Reply: Thank you for your suggestions. I am very sorry for the vague description of pH values of proteins in the range 2-12. At first, we considered to adjust the pH values of protein by using phosphate buffer, but we did realize that it was very inappropriate to use phosphate buffer to adjust different pH values in the actual experiment. Different pH solutions were prepared by using 25 mM Tris–HCl buffer containing 150 mM NaCl adjusted to the different pH values (pH 2, 4, 6, 8, 10, and 12) with the addition of small amounts of hydrochloric acid or sodium hydroxide through a Mettler Toledo FE28-Standard pH meter according to the previously reported method [2]. The DzFer sample was diluted to approx. 0.5 mg/mL with binding buffer, and incubated overnight in different pH buffers. The related contents had been corrected in page 3, line 131-137, and the reference had been added in page 25, line 824-825 in the revised manuscript.

  1. p-5, l-242: “.. oxidized iron content..”. What you mean is Fe(II) content. Or do you suggest that ferrozine also reacts with Fe(III)? p-7, Fig 1A Please explain why the purified ferritin in this gel has an apparent molecular mass of twice (ca 40 kDa) the expected value (20 kDa).

Reply: Thank you for your suggestions. Ferrozine is regarded as a colorimetric reagent and can form the Fe2+-ferrozine complex with free ferrous iron in solution, thus resulting in a violet-colored solution, the absorbance of which can be detected at 560 nm [3]. Thus, in this study, we determined the oxidation reactions of protein samples toward Fe2+ by adding the ferrozine iron reagent and reading the absorbance of the solution at different time points, e.g., 0, 15, 30, 60, 90, and 120 s at 560 nm. In addition, I am very sorry for the mistake about the omission of 25 kDa protein band in protein markers (e.g., 14.4, 18.4, 25, 35, 45, 66.2, 116 and 120 kDa) (Figure 1A, C), and the mistake had been corrected in the revised manuscript. SDS-PAGE analyses defined a SUMO-tagged DzFer subunit as having a molecular weight (MW) of approx. 30 kDa (Figure 1A). After removing the SUMO tag and further purification, the MW of the AjFER subunit was estimated as approx. 20 kDa (Figure 1B, C). The sentences had been added in page 6, line 293-296 in the revised manuscript.

  1. p-9, l-355-etc: I find the results of the Energy Dispersive X-ray Spectroscopy disappointing and not convincing. I suggest the authors to consider removing these results from the paper (or, alternatively move them to Supplementary Materials, but then with a clear explanation of what they think they have measured here). Fig. 4 suggests that the samples contain 30-35 times more Al and Si than Fe. Where do the Al and Si come from? Also, the very small Fe peaks are hardly visible; the Cu and Ag peaks are invisible to me. What have you measured here: metals on the surface? And what is the relation to the rest of the paper, which is about metal ions bound inside the protein?

Reply: Thank you for your suggestions. In this study, before EDS analysis by using SEM-EDS, the protein samples were first fixed to the surface of mica sheets, and then subjected to spray-gold (metallization). Thus, it is speculated that the relatively high abundance ratios of Al and Si elements have been detected in the samples, which is most likely to come from mica sheets, similarly as previously reported [4]. As you suggested, although the surfaces of DzFer, Ag+-DzFer, and Cu2+-DzFer were detected to contain the Fe element, the atomic proportion of which was extremely low. Similarly, the surfaces of Ag+-DzFer and Cu2+-DzFer were found to contain the lower proportions of the Ag and Cu elements, respectively. Collectively, we speculate that the reasons resulting in these problems that are most probably caused by the low protein concentrations in our samples. Thus, we do realize that the EDS results are indeed disappointing and not convincing, and they have been removed in the revised manuscript.

  1. p-9, l-362: The authors find ca 56 Fe atoms per cage. It is worthwhile, here or elsewhere in the paper, to specify that the authors use native ferritin which has not been loaded with Fe in vitro, and that 56 Fe probably means that all ferroxidase centers are filled (48 Fe) and that only a very small amount of Fe may actually be stored in the cage.

Reply: Thank you for your suggestions. In this study, we used the prokaryotic expression system to heterologously express and purify the recombinant DzFer according to the reported method [5]. Notably, the color of the purified DzFer presented yellow (5 mg/mL) and light (1 mg/mL), respectively (Figure S1A, B), suggesting that holo DzFer may be rich in iron. As expected, it was found that the iron content of the purified DzFer molecule contained at 55.68 ± 0.49 atoms per cage as determined by ICP-MS, although less than the number of 90 ± 2 atoms per cage in both HuHF and ChF as reported by De Meulenaere et al. [6]. The related contents had been corrected in page 3, line 113-114, and the solution images of DzFer had been added in Supplementary material Figure S1 in the revised manuscript.

  1. P-9, l-365: “silver ions per monomer” and “copper ions per monomer” should be ‘per cage’, as correctly given in the Discussion (p-18, l-593).

Reply: Thank you for your suggestions. The definition of “per monomer” has been replaced by “per cage” in the revised manuscript.

  1. P-10, l-379: The reported Kd values reported here and elsewhere in the paper require some extra comment from the authors. A single, unique value is reported (the model used for the calculation indeed assumes a single binding site [p-6, l-271), although the subsequent crystal structure analysis identifies five different binding sites both for Cu(II) and for Ag(I). Those sites differ in coordination geometry, and especially sites involving sulfur can be expected to bind much stronger than those involving only oxygen ligands. The authors should warn the reader that the Kd’s are ‘overall’ values and cannot be contributed to specific binding sites, and therefore that the claim of ‘accurate measurement of these interactions’ (Discussion p 18, l-599) should be taken with a grain of salt.

Reply: Thank you for your suggestions. We are verry sorry that the sentence “single-site binding model” is really improper and confusing and has been deleted in the revised manuscript. In the present study, the binding affinity of the interactions of DzFer with Ag+ and Cu2+ was evaluated using MST experiments (Figure 4C, D). We found that both Ag+ and Cu2+ had a high affinity for DzFer, with Kd values of 3.35 ± 0.65 and 8.35 ± 2.85 μM, respectively. Obviously, MST results revealed that there was a much stronger binding reaction for Ag+ with DzFer as compared with Cu2+. However, it is still insufficient for MST analysis to measure the binding affinity under equilibrium condition such as isothermal titration calorimetry (ITC) and identify the specific metal-binding sites. Nonetheless, the binding reaction is definitely not a nonspecific electron static interaction, because this point can be best demonstrated by the results of Cu2+/Ag+ coordination in the crystal structure of Ag+-DzFer and Cu2+-DzFer. The related contents had been corrected in page 22, line 648-659 in the revised manuscript.

  1. p-11, l-407: The authors identify 88 Ag in the crystal structure of Ag+-DzFer; they also identify 5 binding sites per subunit, which would add up to 5 x 24 = 120 Ag. Please comment on the discrepancy between 88 and 120. For Cu (p-11, l-415) the discrepancy is between 56 and 120.

Reply: Thank you for your suggestions. For the Ag+-DzFer structure, one Ag+-DzFer subunit contained five Ag+-binding sites, in which three Ag+ ions were observed to directly coordinate with the amino acid residues of protein subunit. Significantly, two conserved carboxylate residues (Asp127 and Glu130) generated the two symmetry-related metal-binding sites within each three-fold channel, and they were involved in coordination with another two Ag+ ions. Considering the octahedral (432) symmetry of the ferritin shell, the structure of Ag+-DzFer cage was estimated to be bound 88 Ag+ ions. Similarly, there were 56 Cu2+ ions bound to the DzFer nanocage by metal-coordination bonds. Among them, one Cu2+-DzFer subunit also contained five Cu2+-binding sites, in which only one Cu2+ ion was observed to directly coordinate with the amino acid residues of protein subunit. Another four Cu2+ ions were located at the center axis of the three-fold channel and coordinated by six carboxylate groups of the three symmetry-related Asp127 and Glu130 residues.

  1. p-13, l-441: “.. an argentophilic interaction .. was observed”. As far as I can tell, the authors have only observed an Ag-Ag distance of 2.5 Å, which might perhaps suggest the possibility of an argentophilic interaction. Please comment and add a literature reference to ‘argentophilic interaction’.

Reply: Thank you for your suggestions. In the Ag+-DzFer structure, although few studies regarding ferritin with structural data to support this, we observed an argentophilic interaction between Ag1 and Ag2 (the Ag-Ag distance of 2.5 Å) [7], and they coordinated to the negatively charged residues Asp127 and Glu130 at the three-fold channel (Figure 6A, B). The sentences had been added in page 22, line 674-679, and the reference had been added in page 27, line 905 in the revised manuscript.

  1. p-16, Fig 9B: What is the meaning of the broad colored bands surrounding the progress-curve traces?

Reply: Thank you for your suggestions. In order to test the oxidation reactions of protein samples toward Fe2+, the progress curves of the oxidation of Fe2+ into Fe3+ ion related by BSA, DzFer, Ag+-DzFer, and Cu2+-DzFer were monitored at 310 nm at room temperature. Solid lines indicate the average of three technical replicates, and shaded areas represent standard deviation (SD) from the mean. The related contents had been corrected in page 20, line 564-567 in the revised manuscript.

References

  1. Li, Z.; Li, Y.; Zhou, J.; Chundan, Z.; Su, X.; Li, T. A Ferritin from Dendrorhynchus Zhejiangensis With Heavy Metals Detoxification Activity. PloS one 2012, 7, e51428.
  2. Kim, M.; Rho, Y.; Jin, K. S.; Ahn, B.; Jung, S.; Kim, H.; Ree, M. pH-Dependent Structures of Ferritin and Apoferritin in Solution: Disassembly and Reassembly. Biomacromolecules 2011, 12, 1629-1640.
  3. Oliveira, F.; Castro da Costa, A.; Procopio, V.; Garcia, W.; Araujo, J.; Silva, R.; Junqueira-Kipnis, A. P.; Kipnis, A. Mycobacterium abscessus subsp. massiliense mycma_0076 and mycma_0077 Genes Code for Ferritins That Are Modulated by Iron Concentration. Frontiers in Microbiology 2018, 09, 1072.
  4. Chen, L.; Zhou, J.; Zhang, Y.; Chu, S.; He, W.; Li, Y.; Su, X. Preparation and representation of recombinant Mn-ferritin flower-like spherical aggregates from marine invertebrates. PLoS One 2015, 10, e0119427.
  5. Huan, H.; Jiang, Q.; Wu, Y.; Qiu, X.; Lu, C.; Su, C.; Zhou, J.; Li, Y.; Ming, T.; Su, X. Structure determination of ferritin from Dendrorhynchus zhejiangensis. Biochemical and Biophysical Research Communications 2020, 531, 195–202.
  6. De Meulenaere, E.; Bailey, J. B.; Tezcan, F. A.; Deheyn, D. D. First biochemical and crystallographic characterization of a fast-performing ferritin from a marine invertebrate. Biochem J 2017, 474, 4193–4206.
  7. Schmidbaur, H.; Schier, A. Argentophilic interactions. Angew Chem Int Ed Engl 2014, 54, 746-84.

Reviewer 2 Report

The manuscript

Structural and biochemical characterization of silver/copper binding by Dendrorhynchus zhejiangensis ferritin

by Huo et al, submitted to Polymers for publication is a comprehensive work on a ferritin of a marine invertebrate with the focus on its Ag(I) and Cu(II) binding. I think it will be of great interest for the readers of the Journal.

I have few suggestions on the way to improving this manuscript:

1. The authors need to clarify that in Fig.1 A lane 4 the band is related to the purified affinity-tagged DzFer instead of the DzFer itself.

2. Figure 3. The authors state that "As seen from the above results obtained from the CD spectrum, the binding of Ag+ and Cu2+ to DzFer caused no obvious changes to the secondary structures as compared to those of DzFer (Figure 3B)." However, the intensity (molar ellipticity) of the CD spectrum of Ag+-DzFer is half of that of the other two spectra. The spectrum of Cu2+-DzFer is almost identical to that of DzFer. These spectra also do not support the small changes in the secondary structure discussed in the text: "α-helix contents of Ag+-DzFer and Cu2+-DzFer decreased by 17% and 4.2%, respectively, and the β-turn contents increased by 16.9% and 4.1%, respectively, relative to those of DzFer (Figure 3B and Table S3)." In contrast, the intensity of the CD spectrum of Ag+-DzFer in Figure 3B is comparable to those of the DzFer spectra recorded at pH 2 and 12, for which the author calculated drastic decrease of α-helix content – which seems to be reasonable based on the spectral shapes and intensities. Unfortunately, I did not find the supplementary material, to read the data in Table S3. Nevertheless, I suggest to provide the fitting of the CD spectra as a supplementary figure.

3. The CD spectrum of Ag+-DzFer in Figure 3B is very distinct from the others in the wavelength region below 190 nm. This might be caused by high HT during the measurement in this region. The authors shall confirm that the absorbance of this solution was still below the accepted level for proper measurement in this region, or this part of the spectrum needs to be deleted, and should not be considered in the calculations, as well.

4. The absorbance values in Figure 5B in the wavelength region between 200 and 250 nm are too high to be accurate, what is also reflected in the shape of the peaks. A serial dilution of the samples need to be applied to check, whether the concentration dependence is in the linear range. This may also affect the discussion part: " The UV–vis spectrum of Ag+-DzFer showed a slight red-shifted peak of 11 nm (from 211 to 222 nm) compared with the DzFer peak, whereas the optical absorption peak of Cu2+-DzFer also exhibited a corresponding red shift of 24 nm (from 211 to 235 nm) as compared with that of DzFer."

5. The authors provide Kd values for Ag+ and Cu2+ binding of DzFer protein. However, I can not understand, to which process these data are related to, since there are 14 Fe + 26 Ag, or 17 Fe + 113 Cu ions in a monomeric unit. What is the meaning of the "single-site binding model" in this case? I wonder if it is straightforward to provide an average Kd value considering that all metal ions are bound with the same affinity independently of the presence or absence of Fe and of the number of the bound Ag or Cu ions. Also the data in Figs. 9D-9E demonstrate that the metal ions affect each others' binding. The fitting of the MST curves may be ambiguous if there are no experimental point close to the endpoint of the titration.

6. Determining the number of metal ions bound per monomer I would refrain from using significant digits applied by authors: " There were 26.01 ± 0.12 bound silver ions per monomer for the Ag+-DzFer cage as compared to 113.26 ± 0.37 bound copper ions per monomer for the Cu2+-DzFer cage." 0.xx metal ion does not have a physical meaning.

7. Is there any proof against (partial) reduction of bound Cu ions? Especially those Cu2+ ions interacting with Cys residues may undergo reduction to Cu+.

8. The crystal structures of DzFer, Ag+-DzFer and Cu2+-DzFer do not differ significantly in contradiction with the CD spectroscopic data (Figure 3B). The secondary structure composition can also be calculated from the crystal structures. I suggest to include these data for comparison with those obtained from CD in Table S3.

Author Response

Response to Reviewer #2 Comments (highlight in yellow)

  1. The authors need to clarify that in Fig.1 A lane 4 the band is related to the purified affinity-tagged DzFer instead of the DzFer itself.

Reply: Thank you for your suggestions. As you suggested, the note regarding the band of lane 4 in Fig.1 A had been replaced by “purified recombinant DzFer with the SUMO tag” in page 9, line 320-321 in the revised manuscript.

  1. Figure 3. The authors state that "As seen from the above results obtained from the CD spectrum, the binding of Ag+ and Cu2+ to DzFer caused no obvious changes to the secondary structures as compared to those of DzFer (Figure 3B)." However, the intensity (molar ellipticity) of the CD spectrum of Ag+-DzFer is half of that of the other two spectra. The spectrum of Cu2+-DzFer is almost identical to that of DzFer. These spectra also do not support the small changes in the secondary structure discussed in the text: "α-helix contents of Ag+-DzFer and Cu2+-DzFer decreased by 17% and 4.2%, respectively, and the β-turn contents increased by 16.9% and 4.1%, respectively, relative to those of DzFer (Figure 3B and Table S3)." In contrast, the intensity of the CD spectrum of Ag+-DzFer in Figure 3B is comparable to those of the DzFer spectra recorded at pH 2 and 12, for which the author calculated drastic decrease of α-helix content–which seems to be reasonable based on the spectral shapes and intensities. Unfortunately, I did not find the supplementary material, to read the data in Table S3. Nevertheless, I suggest to provide the fitting of the CD spectra as a supplementary figure.

Reply: Thank you for your suggestions. We did realize that the claim that the binding of Ag+ and Cu2+ to DzFer caused no obvious changes of the CD spectrum was being pushed too hard here and should be edited. As depicted in Figure S4A, the CD spectrum of Cu2+-DzFer was nearly overlapped with that of DzFer alone as compared with Ag+-DzFer, indicating that protein secondary structure remained almost unchanged after Cu2+ bound to the DzFer nanocage. It was evident that the α-helix contents of Ag+-DzFer and Cu2+-DzFer were decreased by 17% and 4.2%, respectively, and the β-turn contents were increased by 16.9% and 4.1%, respectively, relative to those of DzFer (Table S3). The structural alteration observed by CD spectroscopy was then accompanied by DLS analyses (Figure S4B). After the binding of Ag+/Cu2+ to the DzFer cage, most Ag+-DzFer and Cu2+-DzFer cages still exhibited a monodispersed distribution with a hydrodynamic size centered at ~13.0 nm, suggesting that protein aggregation hardly occurred in the solution. The pictures regarding the CD spectra and DLS analyses of Ag+-DzFer and Cu2+-DzFer had been moved to supplementary figure S4A, B. The related contents had been reworded in page 11, line 357-367 in the revised manuscript.

  1. The CD spectrum of Ag+-DzFer in Figure 3B is very distinct from the others in the wavelength region below 190 nm. This might be caused by high HT during the measurement in this region. The authors shall confirm that the absorbance of this solution was still below the accepted level for proper measurement in this region, or this part of the spectrum needs to be deleted, and should not be considered in the calculations, as well.

Reply: Thank you for your suggestions. In this study, we performed the circular dichroism (CD) analysis a Jasco J-1500 CD Spectropolarimeter (JASCO Corp., Tokyo, Japan) in the wavelength range from 190 nm to 260 nm at 25 °C as decsibed by the procedure of 2.5.2 in the section of Material and Methods. The DzFer, Ag+-DzFer, and Cu2+-DzFer (approx. 0.2 mg/mL) were respectively scanned at a wide UV range (190–260 nm) with three replicates. As you suggested, although the wavelengths at which the high-tension (HT) voltage exceeded 700 V were excluded during the measurement, the CD spectrum of Ag+-DzFer was extemely distinct from that of Ag+-DzFer, and Cu2+-DzFer. Collectively, we do realize that there may be some problems in actual measurement. On the one side, we overlooked the fact that buffer backgrounds were collected and subtracted from each experiment; On the another side, the HT during the measurement might be still too high as well as an inappropriate protein concentration. Thank you very much for your kindly reminding. We will learn from these experiences and try to avoid these problems in the follow-up measurement process.

  1. The absorbance values in Figure 5B in the wavelength region between 200 and 250 nm are too high to be accurate, what is also reflected in the shape of the peaks. A serial dilution of the samples need to be applied to check, whether the concentration dependence is in the linear range. This may also affect the discussion part: " The UV–vis spectrum of Ag+-DzFer showed a slight red-shifted peak of 11 nm (from 211 to 222 nm) compared with the DzFer peak, whereas the optical absorption peak of Cu2+-DzFer also exhibited a corresponding red shift of 24 nm (from 211 to 235 nm) as compared with that of DzFer."

Reply: Thank you for your suggestions. We realize that it is doubtful that the absorbance values of the DzFer, Ag+-DzFer and Cu2+-DzFer are shown in the wavelength range from 200 nm to 250 nm in Figure 4B. After checking, we very much regret that there were some errors in the process of selecting raw data to profile the diagram of curves. As you suggested, it's really a good proposal for adding a serial dilution of the samples to investigate whether the concentration dependence is within the linear range. We will take full consideration for this proposal in the relevant experiments. Thank you very much for your kindly reminding. The images in Figure 4B had been corrected in the revised manuscript. Obviously, the absorbance spectra of DzFer, Ag+-DzFer, and Cu2+-DzFer all had two maximal protein characteristic absorption peaks. The UV−vis absorption peaks of Ag+-DzFer and Cu2+-DzFer were found at 211/273 and 210/277 nm, respectively, whereas DzFer exhibited two more well-defined spectral peaks at 222 and 279 nm, similar to the spectrum of Tegillarca granosa ferritin (TgFer) [1]. The absorbance spectrum of Ag+-DzFer showed two slight blue-shifted peaks of 11 nm (from 222 to 211 nm) and 6 nm (from 279 to 273 nm) compared with the spectrum of DzFer, while the optical absorption peaks of Cu2+-DzFer also exhibited two corresponding blue shifts of 12 nm (from 222 to 210 nm) and 2 nm (from 279 to 277 nm) as compared with that of DzFer. The sentences had been reworded in page 13, line 399-408 in the revised manuscript.

  1. The authors provide Kd values for Ag+ and Cu2+ binding of DzFer protein. However, I can not understand, to which process these data are related to, since there are 14 Fe + 26 Ag, or 17 Fe + 113 Cu ions in a monomeric unit. What is the meaning of the "single-site binding model" in this case? I wonder if it is straightforward to provide an average Kd value considering that all metal ions are bound with the same affinity independently of the presence or absence of Fe and of the number of the bound Ag or Cu ions. Also the data in Figs. 9D-9E demonstrate that the metal ions affect each others' binding. The fitting of the MST curves may be ambiguous if there are no experimental point close to the endpoint of the titration.

Reply: Thank you for your suggestions. We are verry sorry that the sentence “single-site binding model” is really improper and confusing and has been deleted in the revised manuscript. It is well kown that Fe2+ ions enter into the ferritin cage from the outside environment through 3-fold channels with the help of the carboxylate groups of Glu and Asp residues, and then transferred to the ferroxidase site by the specific array of carboxylate groups from Asp and Glu residues [2]. Except for iron ions, many different types of metal ions, are capable of binding to ferritins, which can potentially affect the rate of catalysis of Fe2+ oxidation in the ferroxidase center [1, 2]. For this reason, it was necessary in this study to supplement the monitoring of the oxidation reaction rates of Fe2+ ions catalyzed by Ag+-DzFer and Cu2+-DzFer. Based on the experiments of iron oxidation activities, the ferroxidase activities of Ag+-DzFer and Cu2+-DzFer was severely affected as compared with native DzFer, which was mostly attributed to Ag+ or Cu2+ binding in the DzFer cage disturbing the interaction of these polymers with Fe2+ ions. As a biophysical assay, MST analysis can sensitively measure molecular interactions between target proteins and metal ions in solution [3]. More specifically, the amount of fluorescence decrease at the heated spot (the MST signal) was altered in the presence of metal ions, thus providing a direct readout of metal ion binding to the proteins [4]. To verify the hypothesis related to these interactions, an MST analysis was performed to assess the binding affinity between Fe2+ and Ag+-DzFer and Cu2+-DzFer. As you suggested, it is still insufficient for MST analysis to measure the binding affinity under equilibrium condition such as isothermal titration calorimetry (ITC) and identify the specific metal-binding sites. We will continue to explore the related research content in the future work. The sentences had been added in page 20, line 573-578, and page 22, line 654-656 in the revised manuscript.

  1. Determining the number of metal ions bound per monomer I would refrain from using significant digits applied by authors: " There were 26.01 ± 0.12 bound silver ions per monomer for the Ag+-DzFer cage as compared to 113.26 ± 0.37 bound copper ions per monomer for the Cu2+-DzFer cage." 0.xx metal ion does not have a physical meaning.

Reply: Thank you for your suggestions. The sentences had been replaced by “The iron content of the purified DzFer was determined to be 56 atoms per cage. In contrast, the binding of Ag+ or Cu2+ to the DzFer cage resulted in significant decreases in the iron contents with 14 and 17 atoms per cage, respectively. There were 26 bound silver ions per cage for the Ag+-DzFer cage as compared to 113 bound copper ions per cage for the Cu2+-DzFer cage.” in page 12, line 393-398 in the revised manuscript.

  1. Is there any proof against (partial) reduction of bound Cu ions? Especially those Cu2+ ions interacting with Cys residues may undergo reduction to Cu+.

Reply: Thank you for your suggestions. We are verry sorry that we just don't know whether Cu2+ion in ferritin can be reduced to Cu+ after interaction with Cys residues according to the available and known literatures. However, it is a good proposal for us to further explore the scientific issues in relevant fields.

  1. The crystal structures of DzFer, Ag+-DzFer and Cu2+-DzFer do not differ significantly in contradiction with the CD spectroscopic data (Figure 3B). The secondary structure composition can also be calculated from the crystal structures. I suggest to include these data for comparison with those obtained from CD in Table S3.

Reply: Thank you for your suggestions. The secondary structure assignments from atomic resolution protein structures were conducted using the STRIDE server (http://webclu.bio.wzw.tum.de/stride/) [5]. The percentages of the secondary structure assignments based on the known crystal structures of DzFer (PDB ID: 7EMK), Ag+-DzFer (PDB ID: 8GY1), and Cu2+-DzFer (PDB ID: 8HCT) had been shown in Figure S5B. CD spectroscopy was applied to obtain more information concerning the changes in the secondary structures of DzFer, Ag+-DzFer, and Cu2+-DzFe (Figure S4A). As compared with DzFer, we observed a significant decrease in the α-helix content in Ag+-DzFer (approx. 17%) and Cu2+-DzFer (approx. 4.2%) (Table S3). Nonetheless, whether in DzFer or in metal-bound DzFer, the content of secondary structures (i.e., α-helix) remained relatively stable based on the calculation by the STRIDE server [5], quite distinct from the results of CD analyses (Figure S5B). Notably, the contents of random coil in Ag+-DzFer and Cu2+-DzFer were increased by 2.3 % and 1.1%, respectively, in comparison with DzFer. This was probably attributed to the interaction of DzFer with Ag+/Cu2+ indirectly affecting the formation of hydrogen bonds and resulting in structural changes in ferritin [6]. The related contents had been corrected in page 5, line 241-243, and page 21-22, line 629-640 in the revised manuscript.

References

  1. Ming, T.; Jiang, Q.; Huo, C.; Huan, H.; Wu, Y.; Su, C.; Qiu, X.; Lu, C.; Zhou, J.; Li, Y.; Han, J.; Zhang, Z.; Su, X. Structural Insights Into the Effects of Interactions With Iron and Copper Ions on Ferritin From the Blood Clam Tegillarca granosa. Front Mol Biosci 2022, 9, 800008.
  2. Maity, B.; Hishikawa, Y.; Lu, D.; Ueno, T. Recent progresses in the accumulation of metal ions into the apo-ferritin cage: Experimental and theoretical perspectives. Polyhedron 2019, 172, 104-111.
  3. Asmari, M.; Ratih, R.; Alhazmi, H. A.; El Deeb, S. Thermophoresis for characterizing biomolecular interaction. Methods (San Diego, Calif.) 2018, 146, 107-119.
  4. Gupta, A. J.; Duhr, S.; Baaske, P. Microscale Thermophoresis (MST). In Encyclopedia of Biophysics, Roberts, G.; Watts, A., Eds. Springer Berlin Heidelberg: Berlin, Heidelberg, 2018; pp 1-5.
  5. Heinig, M.; Frishman, D. STRIDE: a Web server for secondary structure assignment from known atomic coordinates of proteins. Nucleic acids research 2004, 32, W500-2.
  6. Li, H.; Tan, X.; Xia, X.; Zang, J.; El-Seedi, H.; Wang, Z.; Du, M. Improvement of thermal stability of oyster (Crassostrea gigas) ferritin by point mutation. Food Chemistry 2021, 346, 128879.

Round 2

Reviewer 2 Report

1. Answer 1 accepted.

2. Figure 3. The authors did not try to repeat the CD measurement with Ag+-DzFer for which the molar ellipticity is ~ half of that of the other two spectra. If they accept these experimental results and discuss, I suggest to provide a supplementary figure S4C showing the HT values for the three CD spectra as a function of the wavelength. The pathlength of the applied cell shall also be provided in the experimental part of CD measurements.

3. See point 2.

4. The authors repeated the spectrophotometric measurement with the diluted solution of Cu2+-DzFer. The new conditions are not included in the experimental or in the figure. The changes upon dilution clearly show that the discussed differences in the spectra need to be reconsidered. It is clear that the spectrum of DzFer itself also exhibits too high absorbance below 250 nm to be in the linear range. In order to avoid confusion, all the spectra need to be measured accurately at the proper dilution, and the molar absorbance curves shall be compared in Figure 4B.

5. MST might be suitable for determining the Kd values in relatively simple systems. The studied systems here are not such simple systems, since a large number of metal ions is bound to various binding sites probably as clusters. I understand that simplifications have been applied, which need to be mentioned in the text. As far as I understood an average Kd is provided for a hypothetical 1:1 species, i.e M + DzFer = M-DzFer. I only have concern with the fitting of the MST curves, which are ambiguous because there are no experimental points close to the endpoint of the titration – i.e. the S shape of the curve is only indicated by the fit, but not by the experimental data. This means that the data could be fitted equally well with much higher Kd. Thus, the only acceptable information from these fits could be an estimate for the lowest Kd value. This also needs to be considered when presenting such data.

6. Answer 6 accepted.

7. Answer 7 accepted.

8. Answer 8 accepted with the note that it is very unlikely that the secondary structures of e.g. the Cu2+-DzFer after Fe2+ uptake remained the essentially the same as before that (see tables S3 and S4). At the same time the intensity of the CD spectra decreased to half of the original vales during this process (see Figures S4A and 8A). The authors should comment on this. It would be very informative to provide the HT values for all the CD spectra in the supplementary material.

Author Response

Response to Reviewer Comments (highlight in yellow)

  1. Figure 3. The authors did not try to repeat the CD measurement with Ag+-DzFer for which the molar ellipticity is ~ half of that of the other two spectra. If they accept these experimental results and discuss, I suggest to provide a supplementary figure S4C showing the HT values for the three CD spectra as a function of the wavelength. The pathlength of the applied cell shall also be provided in the experimental part of CD measurements.

Reply: Thank you for your suggestions. As you suggested, the images of the HT values during the CD measurements has been shown in Figure S4. In the present study, CD spectra were collected using a Jasco J-1500 CD Spectropolarimeter (JASCO Corp., Tokyo, Japan) in the wavelength range from 190 nm to 260 nm at 25 °C, and the scan rate was 1 nm/min. Samples were scanned three times and then an average was taken with a 1.0 nm bandwidth. All spectra were recorded using a quartz cell with 1 cm path length. The wavelengths at which the high-tension (HT) voltage exceeded 700 V were excluded during the CD measurements. The sentences had been added in page 4, line 163-169 in the revised manuscript.

  1. The authors repeated the spectrophotometric measurement with the diluted solution of Cu2+-DzFer. The new conditions are not included in the experimental or in the figure. The changes upon dilution clearly show that the discussed differences in the spectra need to be reconsidered. It is clear that the spectrum of DzFer itself also exhibits too high absorbance below 250 nm to be in the linear range. In order to avoid confusion, all the spectra need to be measured accurately at the proper dilution, and the molar absorbance curves shall be compared in Figure 4B.

Reply: Thank you for your suggestions. As you suggested, we repeated the spectrophotometric measurements of DzFer, Ag+-DzFer, and Cu2+-DzFer at the concentrations of 3, 1.5, 0.75, and 0.375 mg/mL, respectively. In this tudy, the ultraviolet–visible (UV–vis) absorption spectral measurements were performed using an Epoch 2 microplate spectrophotometer (BioTek Instruments, Inc., Winooski, VT, USA) in the range of 200–600 nm. The experimental conditions were as follows: bandwidth 1.0 nm, scanning speed 100 nm/min, data pitch 1.0 nm. As shown in Figure 4B, the UV−vis spectra were used to analyze the optical absorption characteristics of these proteins. Obviously, the absorbance spectra of Ag+-DzFer and Cu2+-DzFer all had only one maximal protein characteristic absorption peak in the range 260–290 nm, most of which were nearly overlapped to that of DzFer. The maximal absorption peaks of them were mainly found at 276 and 277 nm, except for a 275 nm absorption peak merely observed in Cu2+-DzFer. Although it is reported that the Ag+/Cu2+ can be bound to the ferritin nanocage via the interactions with the Cys and/or His residues [1, 2], the results that the UV−vis spectra of both Ag+-DzFer and Cu2+-DzFer almost overlap with that of DzFer are not sufficient to illustrate the existence of such interactions (Figure 4B). The related contents had been added in page 4, line 187-194, page 10, line 416-422, and page 21, line 676-680 in the revised manuscript.

  1. MST might be suitable for determining the Kd values in relatively simple systems. The studied systems here are not such simple systems, since a large number of metal ions is bound to various binding sites probably as clusters. I understand that simplifications have been applied, which need to be mentioned in the text. As far as I understood an average Kd is provided for a hypothetical 1:1 species, i.e M + DzFer = M-DzFer. I only have concern with the fitting of the MST curves, which are ambiguous because there are no experimental points close to the endpoint of the titration–i.e. the S shape of the curve is only indicated by the fit, but not by the experimental data. This means that the data could be fitted equally well with much higher Kd. Thus, the only acceptable information from these fits could be an estimate for the lowest Kd value. This also needs to be considered when presenting such data.

Reply: Thank you for your suggestions. We also fully agree with your suggestion. As well known, MST is as a powerful technique to perform the qualitative binding studies for molecular interactions based on thermophoresis [3]. MST principle depends on the movement of molecules in a temperature-gradient which can be used for studying the binding affinity of target molecules such as proteins, peptides, metal ions, and many others [4, 5]. So in recent years, MST has developed into be one of the newest biophysical methods commercially available to characterize biomolecular affinities of various molecules irrespective of the type or size of the target [6, 7]. In the present study, MST experiments were conducted using a NanoTemper Monolith NT.115 system (NanoTemper Technologies, Munich, Germany) to determine the equilibrium dissociation constant (Kd) between DzFer and Ag+ or Cu2+. The apparatus can allow the analysis of up to 16 solutions leading to the determination of detailed Kd in a single run within 30 min. After being labeled with fluorescent dye, a total of 5 μL of labeled protein solution at a concentration of approx. 200 nM was added to the standard treated silicon capillaries (Monolith NT.115 series capillaries; catalog no. MO-K022) with different concentrations of Ag+ (from 0.275 nM to 9 µM) or Cu2+ (from 0.61 nM to 20 µM) as well as ferrous solution (from 0.61 nM to 20 µM). MST measurements were performed in triplicate at medium MST power in the red channel using 20% excitation power at 25 °C. The data analysis was carried out using the MO Affinity Analysis v2.3 software (NanoTemper Technologies) to fit the curve and calculate the value of Kd. As shown in Figure S6, fluorescently labeled DzFer moves from a locally heated region to the outer cold region until a relative steady-state is reached (up to approx. 22 s). Addition of Ag+ or Cu2+ decreased protein thermophoretic mobility and consequently increased the normalized fluorescence. The MST binding curves of DzFer+Ag+ and DzFer+Cu2+ revealed that Ag+ and Cu2+ interact with DzFer at Kd values of 3.35 ± 0.65 µM and 8.55 ± 2.85 µM, respectively (Figure 4C, D). The results suggested that MST results revealed that there were some significant interactions of DzFer with Ag+ or Cu2+, whereas the binding reaction for DzFer with Ag+ was much stronger than that of Cu2+. Moreover, to investigate the the interactions of Fe2+ with Ag+-DzFer and Cu2+-DzFer, MST analyses were performed to assess the binding affinities between Fe2+ and DzFer, Ag+-DzFer, and Cu2+-DzFer. The MST time traces of all scanned solutions showed a successive decline in signal indicating that the thermophoresis of protein samples presented a concentration dependent change, which clearly indicated binding of Fe2+ to DzFer, Ag+-DzFer, and Cu2+-DzFer (Figure S8). The thermophoretic traces demonstrate striking differences between the high and low concentrations in response to thermophoresis. Additionally, the binding curves displayed typical decreasing or increasing sigmoidal shapes (Figure 8D-F). The MST results revealed significant binding interactions of Fe2+ ion to DzFer, Ag+-DzFer, and Cu2+-DzFer with different Kd values. In detail, Fe2+ was able to strongly bind to DzFer with a high efficiency, resulting in the lowest Kd value of 2.12 ± 0.88 µM (Figure 8D). Both Ag+-DzFer and Cu2+-DzFer were observed to bind with relatively high affinities to Fe2+ as expected, with Kd values of 18.0 ± 11.5 µM and 7.58 ± 4.16 µM, respectively (Figure 8E, F). The results implied that the binding of Ag+ to DzFer could have a certain inhibitory effect on the interaction of Fe2+ with DzFer. Collectively, it is still insufficient for MST analysis to measure the binding affinity under equilibrium condition such as isothermal titration calorimetry (ITC) and identify the specific metal-binding sites. Additionally, the error bars shown in the Figure 4C-D and 8D-F represent standard deviation of the mean of three values of f(C) [where f(C) is the fraction bound at a given ligand concentration “C”.] obtained for each concentration. It is noteworthy that the errors are relatively high for DzFer+Cu2+, DzFer+Fe2+ and Cu2+-DzFer+Fe2+, probably due to the purity of the labeled protein samples. Thus, the influence of these factors should be considered much more fuuly in the further experiments. The sentences had been added in page 6, line 264-279, page 11, line 430-444, page 19-20, line 603-615, and page 19-21, line 686-692 in the revised manuscript.

  1. Answer 8 accepted with the note that it is very unlikely that the secondary structures of e.g. the Cu2+-DzFer after Fe2+ uptake remained the essentially the same as before that (see tables S3 and S4). At the same time the intensity of the CD spectra decreased to half of the original vales during this process (see Figures S4A and 8A). The authors should comment on this. It would be very informative to provide the HT values for all the CD spectra in the supplementary material.

Reply: Thank you for your suggestions. We do realize that the secondary structure contents of Cu2+-DzFer before and after and Fe2+ uptake are estimated to be subtle differences that is inconsistent with the CD spectra of Cu2+-DzFer and Cu2+-DzFer+Fe2+ (Table S3 and S4; Figure S5B and 8A). On the one side, we provide the figures related to the HT values of DzFer, Ag+-DzFer and Cu2+-DzFer as shown in Figure S4B. On the other side, each sample was performed with three independent measurements at the RMS value of less than 20% during CD measurement. However, we checked the test data carefully, and no any errors were found in data processing and figures generation. We realize that it is most likely that the striking differences in CD data is probably due to different measurement periods and the poor purity of protein samples. So we suggest that the CD data of DzFer, Ag+-DzFer and Cu2+-DzFer as well as DzFer+Fe2+, Ag+-DzFer+Fe2+ and Cu2+-DzFer+Fe2+ should be re-determined if there is enough time, because the preparation of protein samples need some times. Thank you very much for your kindness and patience.

References

  1. Butts, C. A.; Swift, J.; Kang, S.-g.; Di Costanzo, L.; Christianson, D. W.; Saven, J. G.; Dmochowski, I. J. Directing Noble Metal Ion Chemistry within a Designed Ferritin Protein. Biochemistry 2008, 47, 12729-12739.
  2. Huard, D. J.; Kane, K. M.; Tezcan, F. A. Re-engineering protein interfaces yields copper-inducible ferritin cage assembly. Nat Chem Biol 2013, 9, 169-76.
  3. Gupta, A. J.; Duhr, S.; Baaske, P. Microscale Thermophoresis (MST). In Encyclopedia of Biophysics, Roberts, G.; Watts, A., Eds. Springer Berlin Heidelberg: Berlin, Heidelberg, 2018; pp 1-5.
  4. Jerabek-Willemsen, M.; André, T.; Wanner, R.; Roth, H. M.; Duhr, S.; Baaske, P.; Breitsprecher, D. MicroScale Thermophoresis: Interaction analysis and beyond. J Mol Struct 2014, 1077, 101-113.
  5. Asmari, M.; Michalcová, L.; Alhazmi, H. A.; Glatz, Z.; El Deeb, S. Investigation of deferiprone binding to different essential metal ions using microscale thermophoresis and electrospray ionization mass spectrometry. Microchemical Journal 2018, 137, 98-104.
  6. Asmari, M.; Ratih, R.; Alhazmi, H. A.; El Deeb, S. Thermophoresis for characterizing biomolecular interaction. Methods 2018, 146, 107-119.
  7. El Deeb, S.; Al-Harrasi, A.; Khan, A.; Al-Broumi, M.; Al-Thani, G.; Alomairi, M.; Elumalai, P.; Sayed, R. A.; Ibrahim, A. E. Microscale thermophoresis as a powerful growing analytical technique for the investigation of biomolecular interaction and the determination of binding parameters. Methods and Applications in Fluorescence 2022, 10, 042001.
